# Benidipine impairs innate immunity converting sublethal to lethal infections in a murine model of spotted fever rickettsiosis

Andrés F. Londoño[1,2☯], Jennifer M. Farner[1,3☯], Marlon Dillon[4], Dennis J. Grab[2], Yuri Kim[1], Diana G. Scorpio[4], J. Stephen Dumler[2]*

**1** Henry M. Jackson Foundation for Advancement of Military Medicine, Bethesda, Maryland, United States of America, **2** Department of Pathology, School of Medicine, Uniformed Services University, Bethesda, Maryland, United States of America, **3** Emerging Infectious Disease Graduate Program, School of Medicine, Uniformed Services University, Bethesda, Maryland, United States of America, **4** Vaccine Research Center, National Institutes of Allergy and Infectious Diseases, National Institutes of Health, Bethesda, Maryland, United States of America

☯ These authors contributed equally to this work.
* john.dumler@usuhs.edu

**Data Availability Statement:** All relevant data are within the paper and its Supporting Information files.

## Abstract

Spotted fever group rickettsiae are tick-borne obligate intracellular bacteria that infect micro-vascular endothelial cells. Humans and mammalian infection results in endothelial cell barrier dysfunction and increased vascular permeability. We previously demonstrated that treatment of *Rickettsia parkeri*-infected cells with the calcium channel blocker benidipine significantly delayed vascular barrier permeability. Thus, we hypothesized that benidipine, known to be safe and effective for other clinical processes, could reduce rickettsia-induced vascular permeability *in vivo* in an animal model of spotted fever rickettsiosis. Based on liver, lung and brain vascular FITC-dextran extravasation studies, benidipine did not reliably impact vascular permeability. However, it precipitated a deleterious effect on responses to control sublethal *R. parkeri* infection. Animals treated with benidipine alone had no clinical signs or changes in histopathology and splenic immune cell distributions. Benidipine-treated infected animals had marked increases in tissue and blood bacterial loads, more extensive inflammatory histopathologic injury, and changes in splenic architecture and immune cell distributions potentially reflecting diminished $Ca^{2+}$ signaling, reduced innate immune cell activation, and loss of rickettsial propagation control. Impaired T cell activation by *R. parkeri* antigen in the presence of benidipine was confirmed *in vitro* with the use of NKT cell hybridomas. The unexpected findings stand in stark contrast to recent discussions of the benefits of calcium channel blockers for viral infections and chronic infectious or inflammatory diseases. A role for calcium channel blockers in exacerbation of human rickettsiosis and acute inflammatory infections should be evaluated by a retrospective review of patient's outcomes and medications.

**Funding:** This work was supported in part by grant W81XWH-17-1-0668 from the Congressionally Directed Medical Research Programs (CDMRP) Tick Borne Disease Research Program to JSD and DJG https://cdmrp.health.mil/, and by grant R21AI171791 from the National Institutes of Allergy and Infectious Diseases to JSD. The funders had no role in study design, data collection and analysis, decision to publish, or preparation of the manuscript.

**Competing interests:** I have read the journal's policy and the authors of this manuscript have the following competing interests: Marlon Dillon is employed by the National Institutes of Allergy and Infectious Diseases at the Vaccine Research Center, and at the time the work was performed, DGS was also employed at the National Institutes of Health, Vaccine Research Center. Support was provided in part by the NIAID for this work. Other authors have declared that no conflicts of interest exist.

## Author summary

Spotted fever group rickettsiae infect endothelial cells of humans and animals with spotted fever rickettsiosis. Severe infection is associated with marked blood vessel permeability. We showed in vitro that certain calcium-active drugs, such as calcium channel blockers can stabilize or reverse vascular permeability in vitro. This prompted a preclinical study of the calcium channel blocker benidipine in a spotted fever rickettsiosis mouse model designed to yield 50% deaths. While benidipine did not reliably stabilize vascular permeability in this model, it's use led to a significant dose-dependent increase in animal deaths and a marked loss in the control of rickettsial infections, also associated with a significant reduction in the ability to mount an early innate immune response. Hypothetically, this reflected diminished $Ca^{2+}$ signaling and reduced innate immune cell activation. Thus, we also showed impaired *R. parkeri* antigen T cell activation in the presence of benidipine *in vitro* providing support. These findings are distinct in outcome from recent discussions of calcium channel blocker benefits in viral and chronic infectious or inflammatory diseases. A role for calcium channel blockers in exacerbation of human rickettsiosis and acute inflammatory infections should be evaluated.

## Introduction

Spotted fever group rickettsioses are caused by various *Rickettsia* species distributed worldwide for which disease ranges from mild to life-threatening illness. Despite antibiotic availability, case fatality rates are as high as 5–40% [1]. Spotted fever group rickettsiae (SFGR) are obligate intracellular bacteria that primarily target host microvascular endothelial cells (MECs), resulting in endothelial cell dysfunction and increased vascular permeability due to the disruption of paracellular junctions [2–4]. Fatal complications that result from SFGR-induced vascular permeability include hypotensive shock, cerebral herniation, and respiratory failure, among others [1].

Changes in vascular permeability are critical for homeostasis and calcium ($Ca^{2+}$) is important for cellular function, especially endothelial cell integrity [5–7]. Intracellular $Ca^{2+}$ concentration $[Ca^{2+}]_i$ is tightly regulated by ATP-dependent $Ca^{2+}$ pumps, channels and exchangers [8]. To alter MEC function, the regulation of $[Ca^{2+}]_i$ is compromised with some pathogens before MEC contraction and junctional complex disassembly [9,10]. Activation of $Ca^{2+}$ channels allows entry of extracellular $Ca^{2+}$, and $Ca^{2+}$-signaling network crosstalk amplifies physiological and pathophysiological responses [11]. Increased $[Ca^{2+}]_i$ is the pivotal signal for cytoskeletal retraction and interendothelial junction (IEJ) opening that lead to increased vascular permeability [12]. Yet, $Ca^{2+}$ has protean effects as a signaling molecule in diverse cell types, and as a result, targeting $Ca^{2+}$ for therapeutic applications is potentially problematic.

We and others previously evaluated MEC barrier quality and permeability *in vitro* using electric cell impedance sensing (ECIS) and human brain microvascular endothelial cells infected with *Rickettsia parkeri* (a BSL2 model for spotted fever rickettsiosis) or related BSL3 spotted fever group *Rickettsia* species. These demonstrated a pathogen MOI-dependent decrement in transendothelial cell electrical resistance (TEER), an *in vitro* correlate of vascular permeability [3,4,13]. When treated at the same time with the dihydropyridine class calcium channel blocker benidipine, rickettsia-induced TEER was stabilized in a dose-dependent manner [14,15]. Based on observations that increased $[Ca^{2+}]_i$ underlies vascular permeability via its impact on signaling, cortical actin-myosin contraction, disassembly of paracellular junctions, and proinflammatory gene transcription, we hypothesized that the use of benidipine

could reduce vascular permeability *in vivo* by controlling endothelial cell $[Ca^{2+}]_i$ in turn facilitating marked reductions in morbidity and/or mortality in animal models of spotted fever rickettsiosis, even in the absence of specific anti-rickettsial antimicrobial treatment. The results demonstrate a detectable decrease in vascular permeability but also an unanticipated deleterious effect, likely attributable to diminished $Ca^{2+}$ signaling required for innate immunity, leading to earlier severe clinical signs and greater mortality.

## Materials and methods

### Ethics statement

All animal work was approved by Vaccine Research Center Animal Care and Use Committee (proposal # VRC-19-0819) at the National Institutes of Health/National Institutes of Allergy and Infectious Diseases.

### Bacteria

*Rickettsia parkeri* Atlantic Rainforest (ARF) strain, passage 9, was grown in Vero cells with 5% FBS in MEM medium [16]. *R. parkeri* ARF strain was initially isolated from a tick in Colombia, and is used because it is a BSL-2 pathogen. Human pathogenicity is well-established for *R. parkeri* and when subtyped, this includes the ARF strain [16–18]. When the infection was identified in approximately 90% of cells, the intracellular bacteria were partially purified from Vero cell lysate preparations that had been sonicated in sucrose-phosphate-glutamate buffer solution (218 mM sucrose, 3.76 mM $KH_2PO_4$, 7.2 mM $K_2HPO_4$, and 3.9 mM glutamate [SPG]) and frozen at– 80˚C as previously described [16]. An aliquot from this stock was thawed to quantify viable bacteria as previously described [16]. The *R. parkeri* stock used for this study was determined to have 4.02 x $10^9$ viable bacteria per mL, and this was utilized to calculate the challenge dose of rickettsiae.

### Mice and description of experimental design

To conform to the prior established model [16], eight-week-old male C3H/HeN mice from Charles River Laboratories were used in this study. All animal work was approved by Vaccine Research Center Animal Care and Use Committee (proposal # VRC-19-0819) at the National Institutes of Health/National Institutes of Allergy and Infectious Diseases.

Benidipine HCl was obtained from Sigma and resuspended in dimethyl sulfoxide (DMSO) to a stock concentration of 125 mg/mL. Immediately prior to use, the benidipine was diluted in saline 1:33, 1:100 and 1:333 to working concentrations of 3.75, 1.25, and 0.375 mg/mL, respectively, prior to a 200 μL intraperitoneal (IP) injection. Benidipine doses were based on the 10 mg/kg/d doses previously used for I.P. injection [19]. To optimize treatment, 3 mg/kg/d was used as a lower concentration. The highest dose, 30 mg/kg/d was used since, the $LD_{50}$ values of benidipine in acute toxicity studies were 321.6 and 384.5 mg/kg orally in male and female mice, respectively [20].

Eight groups each containing 12 mice were created. An inoculum of *R. parkeri* calculated to represent an $LD_{50}$ (50% lethal) challenge within the 6 day course of infection was derived from data in Londoño et al. [16], and confirmed in pilot experiments. Thus, cell-free *R. parkeri*, (5 x $10^7$ bacteria in 200μL) were inoculated intravenously (IV) via tail vein into C3H/HeN mice; controls received the same volume of PBS. Mice initially received 0 (vehicle only, 1% DMSO), 3, 10 and 30 mg/kg/d benidipine IP concurrent with rickettsia or PBS IV inoculation. Injection of freshly prepared drug was repeated daily for each day of the experimental protocol. Animals were monitored daily for signs of disease: ruffled fur, erythema, labored breathing, decreased

**Table 1. Primers and probes used in this study.**

| Target organism | Target gene | Sequence (5' → 3') |
|---|---|---|
| *Rickettsia* spotted fever group | *sca0* | Forward TTGTCAGGCTCTGAAGCTAAAC |
| | | Reverse AGCACCTGCCGTTGTGATATC |
| | | Probe FAM-TAGCCGCAGTCCCTACAACACCGC |
| Mouse | *Gapdh* | Forward CAACTACATGGTCTACATGTTC |
| | | Reverse CTCGCTCCTGGAAGATG |
| | | Probe TET-CGGCACAGTCAAGGCCGAGAATGGGAAGC |

activity, hind leg paralysis, and hunched back posture. Body weight was measured until the endpoint day for each group.

The protocol was planned such that half of the animals would be euthanized and necropsied on day 3 and the remainder on day 6; spleen weights were measured at necropsy for all. For fluorescence imaging of vascular permeability, two mice per group were perfused in the tail vein with 10 mg of fluorescein-labeled anionic lysine-fixable 70 kDa dextran (Invitrogen, Waltham, MA) mixed in 200 µL of PBS [21], and the animals were sacrificed after an hour of perfusion. Other tissues obtained at necropsy included liver, lung, and brain. H&E histology, PCR, and immunohistochemistry (IHC) were performed on all tissues to study histopathologic inflammatory injury, rickettsial load, visualization of rickettsial tissue distribution, and identification of immune cells.

## Measurement of *R. parkeri* load by qPCR

DNA was extracted from tissue and blood using the Qiagen DNeasy Blood and Tissue Kit (Qiagen, Valencia, CA), following the manufacturer's protocol. The *sca0* (*ompA*) gene was used to determine *R. parkeri* load and was normalized using mouse *Gapdh* housekeeping gene in tissue or expressed as copies per milliliter of blood (**Table 1**) [22,23]. qPCR was performed in CFX384 Touch Real-Time PCR Detection System (Bio-Rad, Hercules, CA) in 10 µL reaction volumes, which contained 5 µL of KAPA probe fast qPCR Master Mix (2X) Kit (Kapa Biosystems, Wilmington, MA), 0.02 µL from each primer and probe at 100 µM, and 2.94 µL of molecular-grade water, leaving 2 µL for the samples, controls, or standards. The standard curve was prepared with dilutions from $10^9$ to $10^0$ copies/µL of *sca0* and *Gapdh*, and the results were analyzed in CFX Maestro Software version 1.0 (Bio-Rad, Hercules, CA). The bacteria load was expressed as copies of *sca0* per $10^6$ copies of *Gapdh*.

## Histology, immunohistochemistry, and fluorescence imaging

Necropsied tissues for microscopy were fixed with 10% neutral buffered formalin and then embedded in paraffin. The samples were sectioned at 5 µM thickness and stained with hematoxylin and eosin (H&E) for histological analysis. Slides with liver, lung and brain sections were evaluated and ranked from the least pathological features to the most severe. Features considered included presence, density and quality of inflammatory cell infiltrates; specific microanatomical distribution, cellular injury including apoptosis, necrosis, vasculitis and endothelialitis, thrombosis, edema, and hemorrhage. All animals were evaluated for lobular hepatitis, meningitis/encephalitis, and interstitial pneumonitis.

Splenic H&E-stained sections were evaluated with regard to: follicle architecture; follicle activation; white and red pulp volume (as a proportion of total splenic volume); red and white pulp cellularity; quantity of trilinear hematopoietic progenitor cells; inflammatory foci; necrosis; and granuloma-like lesions. For immune cell subset characterization by IHC in splenic and

liver tissues, 5 micrometer-thick sections were deparaffinized and rehydrated, and stained using the Leica Bond system with rabbit primary antibodies against mouse CD3 (all T lymphocytes, Abcam, ref. ab16669, Cambridge, U.K.), CD4 (Abcam, ref. ab237722), CD8 (Abcam, ref. ab217344), NCR1 (NK cells, Abcam, ref. ab233558); in addition, ICAM1 (Abcam, ref. ab179707) was examined in liver tissues. The specific cell immunophenotypes were selected to interrogate the major immune and antimicrobial cells within the spleen with regard to the dynamics of splenic reorganization with infection and effective vs. dysfunctional immune activation. A manual protocol using a rabbit antibody against spotted fever group *Rickettsia* (anti-*Rickettsia conorii* Rc7, a kind gift from Juan Olano, MD, UTMB, Galveston, Texas) was used to detect rickettsial antigen in spleen and liver, and rat primary antibody was used to detect mouse B lymphocytes (CD45R/B220, BD Pharmingen, San Jose, CA) in spleen by a previously described protocol [16].

To correlate cellularity and cellular immunophenotypes in distinct splenic compartments identified by IHC, adjusting for total splenic size, semiquantitative estimates of cellularity were evaluated by comparison with mock-infected, no benidipine treated controls. Control animals were assigned a value of 0, such that increased cellularity would be reflected by positive values and decreased cellularity by negative values (+4 to -4). To accommodate the relationships in spleen size and total cellularity, each semiquantitative cellularity value for H&E and immunophenotype IHCs were divided by the normalized median spleen weight for that group of mice (normalized spleen weight = median of the group/median of the uninfected untreated control for that day).

For fluorescence imaging of FITC-Dextran [21], formalin-fixed paraffin-embedded tissues were sectioned at 5 micrometer thickness. The paraffin was melted at 56˚C for an hour, washed with xylene, rehydrated with alcohol gradients, and the tissue samples were mounted with antifade mounting medium with DAPI (Vector, Burlingame, CA) to be observed by fluorescence microscopy. Vascular leakage of FITC-dextran was measured by image analysis (ImageJ) of a representative field from least two animals per group, calculating the percent FITC fluorescence area in and around surrounding veins, venules and sinusoidal regions within the hepatic lobules, in brain meningeal and parenchymal vessels as well in lung microvessels. Similarly, ICAM-1 expression was quantified from hepatic IHC images of at least two animals per group to determine the integrated density for each. For FITC-Dextran and ICAM-1 IHC measurements, the mean expression of groups was compared using two-sided Student's t-tests, where a p value $< 0.05$ was considered significant.

## Cytokine analysis

Cytokines and chemokines in the plasma samples from all the experimental groups and the two endpoints were detected using the Bio-Plex Pro platform (Bio-Rad, Hercules, CA). A panel of seven cytokines (IL1α, IL2, IL6, IL10, IL12αβ [IL12p70], IFNγ, and TNF [TNFα]) and three chemokines (CXCL1 [KC], CCL2 [MCP-1], and CCL5 [RANTES]) was used and performed as per the manufacturer's protocol, and analyzed using the Bio-Plex 200 system (Bio-Rad, Hercules, CA). The specific cytokines and chemokines were chosen for their capacity to analyze changes in both immune (IFNγ, IL-2, IL-12αβ, IL10) and inflammatory (IL6, TNF) induction, as well to interrogate chemokines involved in lymphocyte and macrophage migration (CXCL1, CCL2, CCL5), as per prior observations and experience with severe inflammatory disease caused by other rickettsiae in situations of disturbed immune signaling ($Stat1^{-/-}$) [24]. For some values, the result was classified as "out of range, high" or "out of range, low"; for these the highest and lowest value, respectively, detected on the standards was assigned, as minimum or maximum value estimates.

## Suppression of innate immune activation by benidipine in vitro

We used NKT cells to study the effect of benidipine on activation by *R. parkeri* soluble antigens. NKT cells are innate immune lymphocytes that express NK cell markers and have T-cell receptors (TCRs). NKT cells are activated by interactions between the TCR and glycolipid antigens that are presented by antigen-presenting cells (APCs). One attribute of NKT cells is cytokine production, an indication of NKT cell activation. L-CD1d cells are APCs derived from the LMTK mouse fibroblast cell line and are transfected to express mouse CD1d. L-vector cells are derived from LMTK mouse fibroblast cells transfected with an empty vector to serve as controls. The L-CD1d and L-vector cells were cultured in DMEM with 10% FBS, 2mM L-glutamine, 0.5mg/mL G-418, and 100 units/mL penicillin/streptomycin [25]. *R. parkeri* antigen was prepared as described for clinical cellular immunity studies of *ex vivo* lymphocytes in a human clinical challenge trial of a Rocky Mountain spotted fever (*R. rickettsii*) vaccine [26]. Briefly, *R. parkeri* was propagated in human brain microvascular endothelial cells and when heavily infected, cells were scraped from the flask and centrifuged at 15,000$g$ for 10 min. The pellet was resuspended in PBS and passed through a 27 gauge needle five times to prepare partially cell-free bacteria. The unbroken or large fragments of cells were removed by centrifugation at 500$g$ for 5 min and the supernatant containing cell-free bacteria was collected and centrifuged at 17,000$g$ for 10 min. The bacterial pellet was resuspended in PBS and sonicated at max power setting for 30 min. After sonication, the protein concentration of this soluble *R. parkeri* fraction was measured using the Pierce BCA Protein Assay Kit (Thermo Scientific).

For immune cell activation studies, L-CD1d cells, that present endogenous glycolipids, and control L-vector cells lacking CD1d expression were pulsed for 16h with 33 μg/mL *R. parkeri* unfractionated protein antigen with and without 10 μM benidipine. Vα14+ TCR invariant NKT (iNKT) N8-3C3 hybridoma cells were cultured in IMDM with 5% FBS, 2mM L-glutamine, and 100 units/mL Penicillin/Streptomycin. After pulsing, the L-CD1d and L-vector cells were washed with 1x PBS and set up in co-culture with N8-3C3 NKT hybridoma cells with and without 10 μM benidipine for 24 hours. The supernatants from the co-cultures were collected and IL-2 levels measured by ELISA (BD biosciences, 555148). The L-CD1d, L-vector, and iNKT cells were graciously provided by Dr. Tonya Webb (University of Maryland School of Medicine, Baltimore, MD).

## Statistical analysis

Data were organized using Microsoft Excel. The GraphPad Prism 9 program and https://astatsa.com were used to analyze differences in groups using parametric (Student's t-test) or non-parametric statistical tests (Kruskal-Wallis test for multiple comparisons and Mann-Whitney test for two groups comparison). P values $<0.05$ were considered significant. For Kruskal-Wallis tests rank sum test for multiple independent samples, if significant at a p-value $<0.05$, pairwise multiple comparison tests were conducted using the Conover post hoc method further adjusted by the Benjamini-Hochberg False Discovery Rate method.

## Results

### Clinical findings

Animals infected with the sublethal dose of *R. parkeri* (5 x $10^7$) but which did not receive benidipine, developed clinical signs and significant weight loss by day 3, and had significant clinical signs, but were alive at day 6 when euthanized for necropsy, similar to that observed previously, approximating the results expected from an estimated $LD_{50}$ challenge dose [16]. Control uninfected animals that did not receive benidipine did not have clinical signs or weight loss at

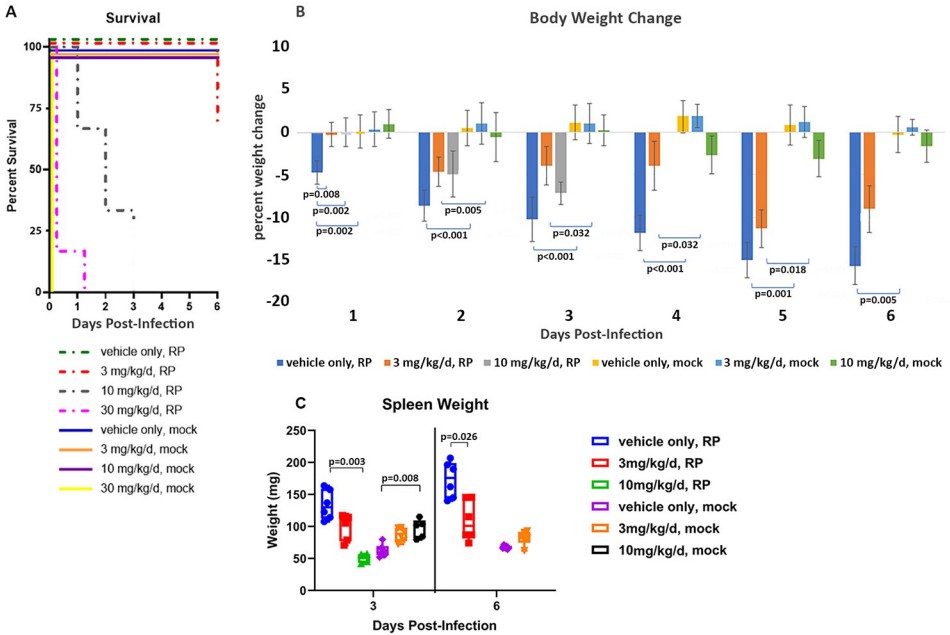

**Fig 1. Kaplan Meier survival curve (A) of mice after challenge with *R. parkeri* (RP) or PBS (mock) and treatment with different benidipine concentrations.** All uninfected mice survived, except all mice treated with 30 mg/kg/d benidipine died within 24h; otherwise, only mice treated with benidipine at 10 mg/kg/d or 3 mg/kg/d and infected by *R. parkeri* died in the 6-day study, including 8 mice treated with 10 mg/kg/d benidipine on day 3 (4 were moribund but euthanized as per protocol), and 2 mice treated with 3 mg/kg/d benidipine on day 6. (B) Changes in weight among all the groups. Bars represent mean weight change and the error bars show the standard deviations. Infection led to marked body mass loss over 6 days that was abrogated on day 3 and 6 by benidipine administration. (C) Spleen weight changes among the groups. Values of all animals are shown as dots. Benidipine caused modest splenomegaly in uninfected mice at 10 mg/kg/d, whereas *R. parkeri* infection led to significant splenomegaly that was abrogated in a dose-dependent manner by benidipine. Median rank is indicated by the line in the box, and the box boundaries show the 1st and 3rd quartiles, whereas the bars show the minimum and maximum ranks.

days 3 or 6; uninfected mice that received 3 or 10 mg/kg/d benidipine survived and also showed no clinical signs.

All animals that received 30 mg/kg/d benidipine, regardless of infection status had an unanticipated adverse reaction that led to death within 30 minutes to overnight (**Fig 1A**). For this reason, the 30 mg/kg/d dose of benidipine and animals that received it were excluded from the protocol. *R. parkeri*-infected animals that received 10 mg/kg/d benidipine had clinical signs of infection by day 1, and 8 were found dead between days 1 and 2 (**Fig 1A**); the remaining 4 animals were moribund on day 3 and were euthanized. Infected animals treated with 3 mg/kg/d had clinical signs by day 3, and significantly less weight loss on day 1 and 6 for the treated groups compared to mice which did not receive benidipine (**Fig 1B**). Two *R. parkeri*-infected mice treated with 3 mg/kg/d benidipine died on day 6 (**Fig 1A**). Treatment of uninfected mice with benidipine led to a slight dose-dependent increase in spleen weight between days 3 and 6 (**Fig 1C**), but infection with *R. parkeri* led to dramatically increased spleen size that was abrogated by benidipine in a dose-dependent manner.

## Histopathology

**H&E staining.** Tissue sections stained with H&E from lung, brain, and liver were evaluated and showed no pathology to extensive histopathologic damage (**Figs 2, 3, and S1–S4**). Since the data were not normally distributed, ranks were used, starting with no pathology

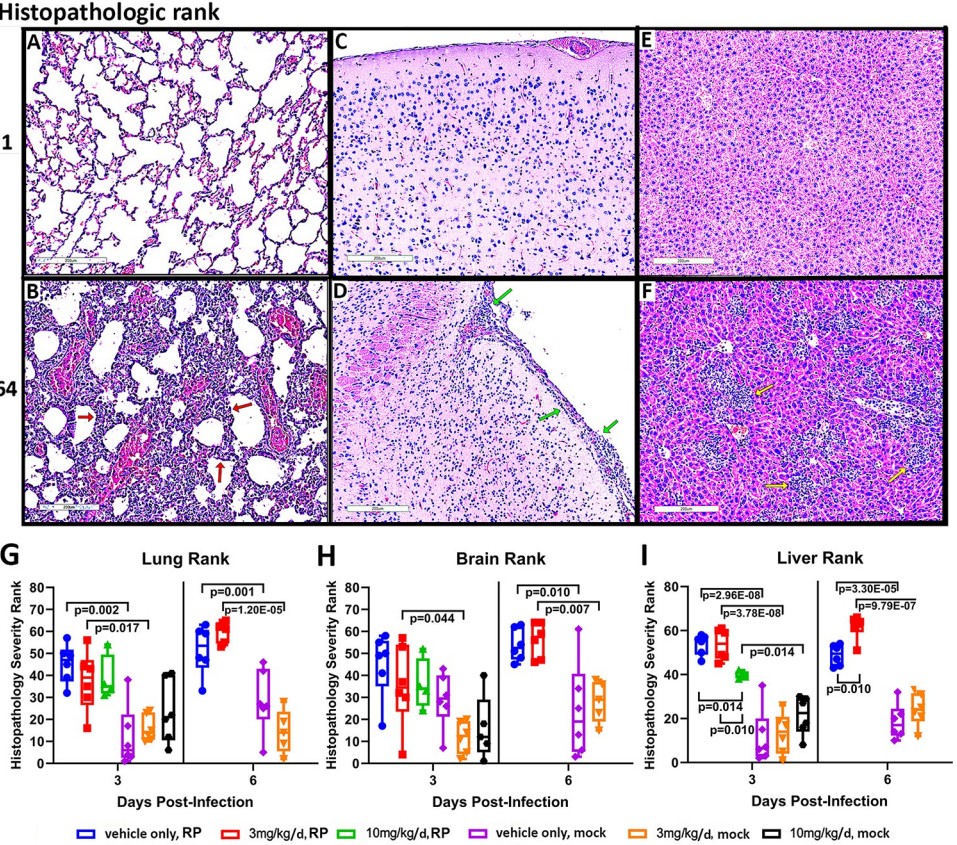

**Fig 2. A-F**. Representative histopathological sections from lung (A-B), brain (C-D), and liver (E-F) ranked from least (rank 1) to greatest (rank 64) histopathologic inflammation and injury among the 64 animals. The top row (A,C,E) shows the pathological findings of those with the lowest rank (rank 1) for lung, brain and liver, and the bottom row (B, D,F) shows those with the highest rank (rank 64) with the most severe histopathological changes. The lowest rank demonstrates the absence of significant inflammatory histopathology and was an attribute of uninfected animals, with or without benidipine treatment. The highest ranks were found exclusively in infected mice. Note the significant interstitial pneumonitis (B) in the lungs (red arrows), the mononuclear cell inflammatory infiltrate in the meninges (D) of the brain (green arrows), and dense hepatic (F) intralobular clusters of mononuclear cell infiltrates (yellow arrows). Bar = 200 μm. **G-I**. ndividual histopathology ranks are shown by markers; median rank is indicated by the line in the box, and the box boundaries show the 1$^{st}$ and 3$^{rd}$ quartiles, whereas the bars show the minimum and maximum ranks.

(lowest rank) to greatest pathology (highest rank) to ascertain significant differences between groups using non-parametric statistical approaches. All non-infected mice had the least histopathologic findings, as measured by the median ranks in lungs, brain and liver. Benidipine treatment with 3 or 10 mg/kg/d did not change histopathologic findings in mock-infected mice. Livers from mock-infected animals had no inflammatory cell infiltration; however, modest hepatocyte vacuolation was observed in mock-infected animals treated with benidipine 10 mg/kg/d. Because of acute toxicity in mice treated with 30 mg/kg/d benidipine, histopathology was not examined in either infected or uninfected animals.

Histopathologic severity increased with infection and was significantly worse for all organs evaluated by day 6. Among infected animals, histopathological features in lungs were characterized by interstitial pneumonitis, vasculitis, capillaritis, necrosis, and nuclear debris (**Fig 2B and S4**), although no significant differences in the presence of these features was found among benidipine-treated and–untreated infected animals.

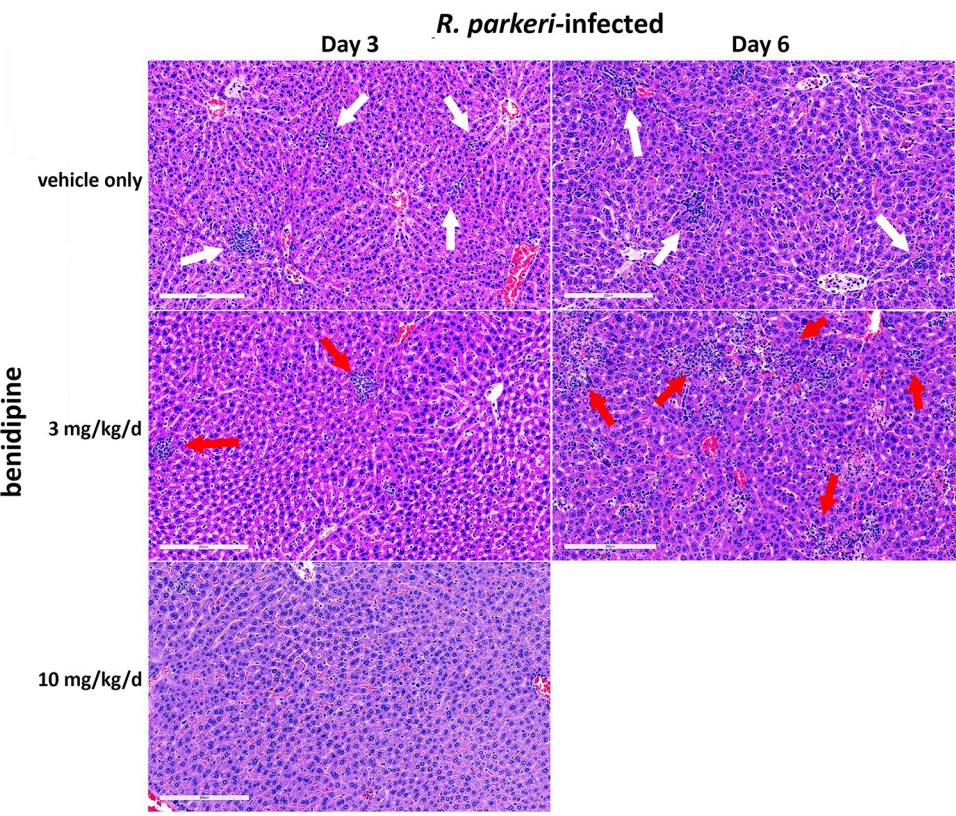

**Fig 3.** *R. parkeri* (RP)-induced hepatic histopathology with/without benidipine (A). Note similar lobular mononuclear cell inflammatory infiltrate histopathology without drug (vehicle only; white arrows) compared to with 3 mg/kg/d (red arrows), but less pathology in those treated with 10 mg/kg/d at day 3. On day 6, animals treated with 3 mg/kg/d benidipine revealed more pathology (red arrows) compared to the group without drug (white arrows). Bar = 200 μm. Also see S1 and S2 Figs for high resolution images and specific histopathologic lesions.

In brains, the main finding was mononuclear cell meningitis (**Fig 2D and 2H**), which was variable in severity and not significantly different among treated and untreated infected animals. Hepatic pathology in infected mice at days 3 and 6 was characterized with hepatic lobular and periportal mononuclear cell infiltrates/hepatitis (**Figs 2F and 3**). Greater degrees of severity included lobular aggregates of mononuclear cells and neutrophils as well as apoptotic cells and necrotic foci, with rare thromboses (**S1 Fig**). Compared to no drug at day 3, the degree of hepatic inflammation was similar in animals treated with 3 mg/kg/d benidipine. However, hepatic pathology on day 3 was quantitatively (p = 0.014; **Fig 2I**) and qualitatively (**Figs 3 and S2**) less severe in those receiving 10 mg/kg/d benidipine compared to no drug or 3 mg/kg/d benidipine. By day 6, hepatic inflammation was significantly greater in infected animals treated with 3 mg/kg/d benidipine compared to no drug (p = 0.010), including increased presence of neutrophils, apoptoses and necrosis in the latter (**Figs 3, S1 and S2**). Thrombosis and aggregates of necroinflammatory debris were more frequently present in infected animals at day 6 treated with 3 mg/kg/d benidipine (**S1 and S2 Figs**). Uninfected animals, whether benidipine-treated or not, had no or inconsequential histopathology (**Fig 2E**) and the lowest severity ranks (**Fig 2I**).

**Dextran-FITC diffusion.** Vascular permeability, as measured by fluorescein-labeled 70kD dextran extravasation (an increase of extravascular/perivascular/perisinusoidal and tissue green fluorescence) was tracked in liver, brain and lung using fluorescence microscopy.

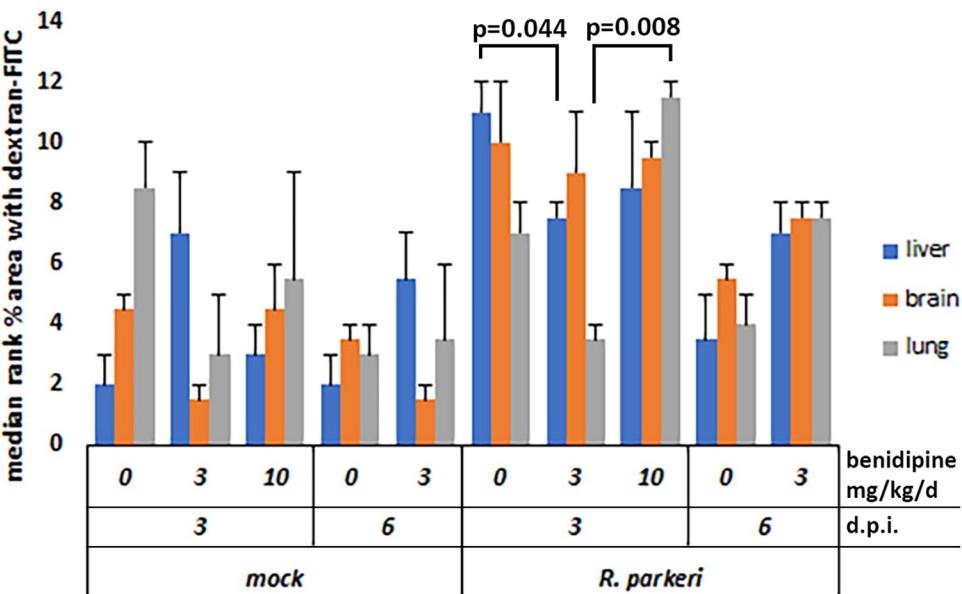

**Fig 4. Measurement of vascular permeability ± benidipine in mock and *R. parkeri*-infected animal tissues.**
Animals were perfused with FITC-Dextran prior to necropsy. While significant differences between mock and *R. parkeri*-infected animals were readily observed, differences were observed between no benidipine and doses calculated to achieve stabilized vascular permeability *in vivo*. Only the liver had a significant reduction in leakage on day 3 p.i. when treated with 3 mg/kg/d benidipine compared to no benidipine in infected animals. Results were calculated as in the methods, and ranked for non-parametric statistical tests.

Only in the liver did benidipine marginally decrease hepatic vascular permeability in uninfected treated mice vs. no drug-treatment, but also marginally suppressed *R. parkeri*-induced vascular permeability with 3 mg/kg/d benidipine compared to no drug (**Fig 4**). This was not observed in infected animals treated with 10 mg/kg/d benidipine, or in the brains of any treated animals suggesting no impact, or at best a very weak impact on tissue vascular permeability with infection. Benidipine had no demonstrable effect on dextran leakage in either brain or lung tissues with *R. parkeri* infection (**S5 Fig**).

**Splenic histopathology and IHC.** Given the observed loss of immune control with benidipine during *R. parkeri* infection, we examined spleen, the immune system tissue that responds predominantly to bloodstream infections, using H&E histopathology and immune cell immunohistochemistry to correlate cellularity, architectural changes, and immune cell phenotype with events, including clinical signs and infection outcome. Among mock-infected animals, spleen weight increased slightly with increasing benidipine doses (**Fig 1C**), and histopathologic changes revealed a modest increase in white pulp and red pulp cellularity over time and benidipine dose (**Figs 5 and S3**). In the absence of infection, white pulp and follicular architecture that reflect effective immune cell maturation and activation were observed and showed slight expansion including minimal follicle fusion in animals treated with 3 mg/kg/d benidipine. Similarly, the red pulp:white pulp volume ratio increased marginally in uninfected mice treated with 10 mg/kg/d benidipine. In contrast, *R. parkeri* infection led not only to marked splenic weight increase, but also to architectural abnormalities of splenic white pulp with follicular dissolution, loss of cellularity in marginal zones and T cell zones/periarteriolar lymphoid sheaths (TCZ/PALS) increasing from day 3 to 6. Changes also included expanded and hypercellular red pulp, where inflammatory foci (<50 μm diameter) and necrosis were readily observed (**Figs 5 and S3**).

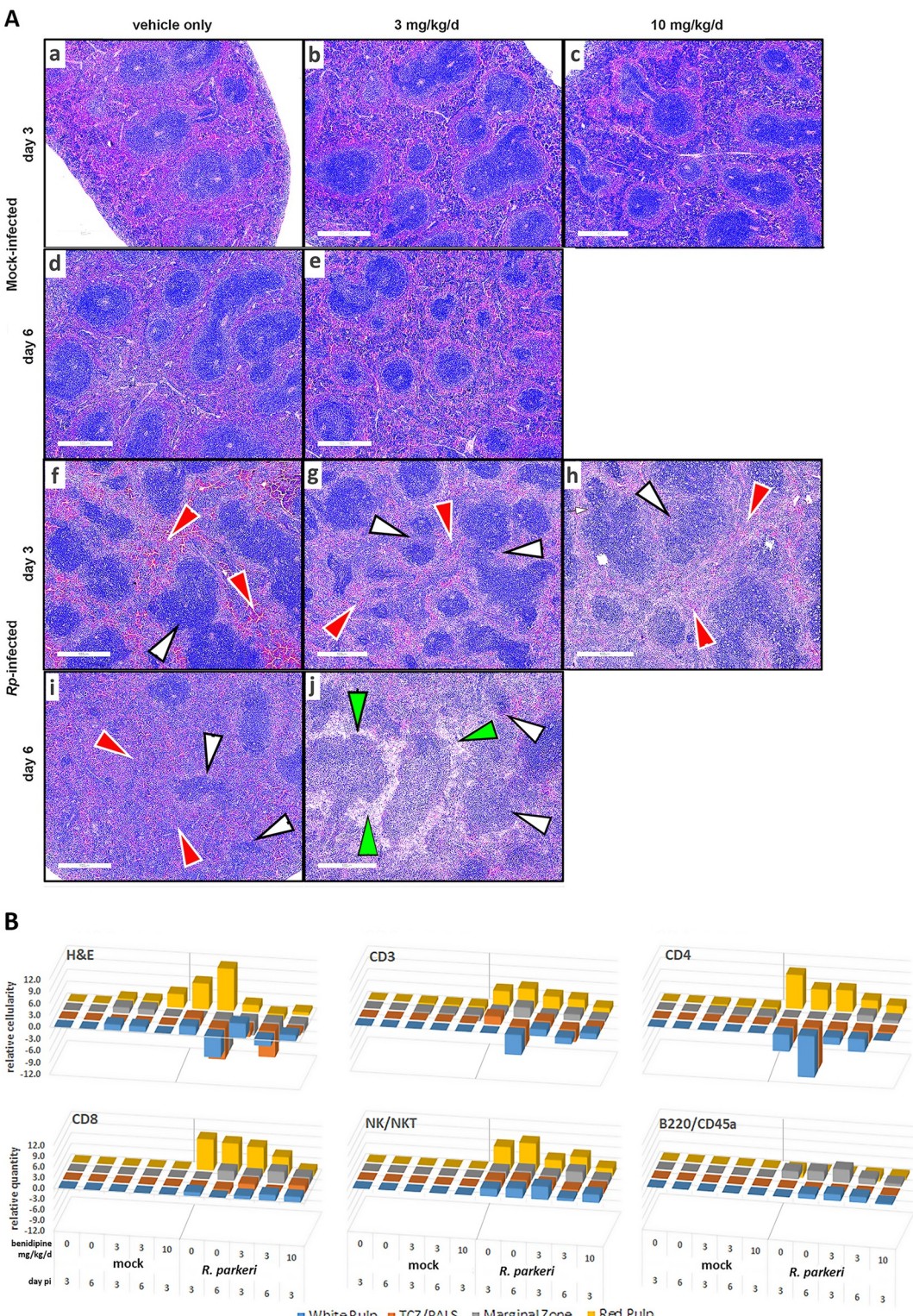

**Fig 5. Histology (A) of splenic immune architecture in response to _R. parkeri_ infection and benidipine treatment (H&E stains; bars = 500 μm).** (a-e) Note the normal follicular structure (white arrows) and maintained red pulp (red arrows) volume ratio despite benidipine treatment in the absence of _R. parkeri_ infection. With infection but no benidipine (f,i), the proportion of splenic volume occupied by white pulp and follicles decreased as a result of increasing red pulp (red arrows) volume and cellularity and fusion/fragmentation of follicles (white arrows). With infection and benidipine treatment (g,h,j),

red pulp expansion and cellularity (red arrows) are minimized while white pulp follicles appear fragmented (white arrows); by day 6, red pulp volume was characterized by increasing cellularity (red arrows) in the absence of benidipine (i) and extensive necrosis (green arrows) was present with benidipine (j); follicular architecture was disorganized and indistinct (white arrows) from red pulp (f-j). (**B**) Panels H&E, CD3 T cells, CD4 T cells, CD8 T cells, NK/NKT cells, and B220/CD45a B cells show the semiquantitative cellularity measurements weighted by spleen size (**Fig 1C**) relative to day 3 mock-infected, no benidipine controls (see methods). Here, benidipine induced little change in immune cell distribution or quantity in the absence of infection. Note the marked relocation and recruitment of CD4/CD8/NK/NKT cells to the red pulp with infection and their loss from white pulp (including TCZ/PALS and marginal zone), partly abrogated by benidipine. B cells were relatively unaffected (see **S4–S8 Figs** for CD4/CD8/NK/NKT/B220 IHC images). TCZ/PALS–T cell zone and periarteriolar lymphoid sheath.

In infected animals, 3 mg/kg/d benidipine-treatment abrogated the loss of white pulp and expansion of red pulp at day 3, but this was overcome by day 6. The chief differences between infected animals with and without benidipine treatment were that spleen weight increased with infection was abrogated with increasing benidipine doses (**Fig 1C**). This corresponded to a marked suppression of red pulp hypercellularity (antimicrobial function) and a marked reduction in cellularity in the TCZ/PALS component of the white pulp that in part governs immune surveillance and activation (**Figs 5A, 5B and S3**). The latter was characterized by accelerated dissolution of white pulp, follicles, marginal zones, and TCZ/PALS associated and extensive red pulp neutrophilic inflammation and necrosis (**Figs 5A and S3**). By day 6, infected, untreated mice had near total loss of white pulp and follicles, with the appearance of small granuloma-like aggregates where neutrophilic infiltrates had collected. In contrast, those treated with benidipine (3 mg/kg/d), had extensive necrosis in both red and white pulp and lacked the presence of the granuloma-like structures (**Figs 5B and S3**).

Immune cell population density and locations changed dramatically over the course of infection, wherein red pulp expansion was the result of increases in CD3, CD4, CD8 and NK/NKT cells (**S6–S10 Figs**), while white pulp contraction with infection was largely the result of CD4 cell loss (**S7 Fig**). Benidipine treatment abrogated but did not eliminate loss or relocation of CD4 cells from the white pulp and TCZ/PALS (**Figs 5 and S7**). Compared to no benidipine in infected animals, 3 mg/kg/d led to modest increases in CD8 cells in all white pulp compartments and modest decreases in the red pulp, whereas 10 mg/k/d benidipine exaggerated both the increase in white pulp and reductions in red pulp CD8 cells (**S8 Fig**). NK and NKT cells (**S9 Fig**) were increased in cellularity in all white pulp regions with 10 mg/kg/d benidipine compared to untreated *R. parkeri*-infected mice. Changes in the distributions of B cells were not significantly impaired by benidipine (**Figs 5 and S10**).

**ICAM1 IHC.** ICAM1 expression in hepatic sinusoids was higher in the infected animals than in mock-infected controls ($p < 0.05$), but unchanged by benidipine treatment (**Fig 6**). Significant differences in expression were not observed with any benidipine dose or between days 3 and 6.

## Tissue rickettsial load by qPCR and immunohistochemistry

All tissue samples collected from infected animals contained rickettsial DNA, and all samples collected from mock-infected animals were negative by qPCR. On day 3, infected animals treated with 10 mg/kg/d benidipine showed higher bacterial loads when compared to untreated animals in all tissues, and mice treated with 3 mg/kg/d benidipine had higher bacterial loads in liver ($p = 0.027$). There were similar results on day 6 for infected animals treated with 3 mg/kg/d benidipine, where more bacteria were found in all tissues excluding blood, when compared to the no drug group (**Fig 7**). There was a benidipine dose-dependent increase in rickettsemia compared to no drug ($6.9 \times 10^2$ *sca0* copies/mL) increasing to $7.9 \times 10^3$ *sca0*

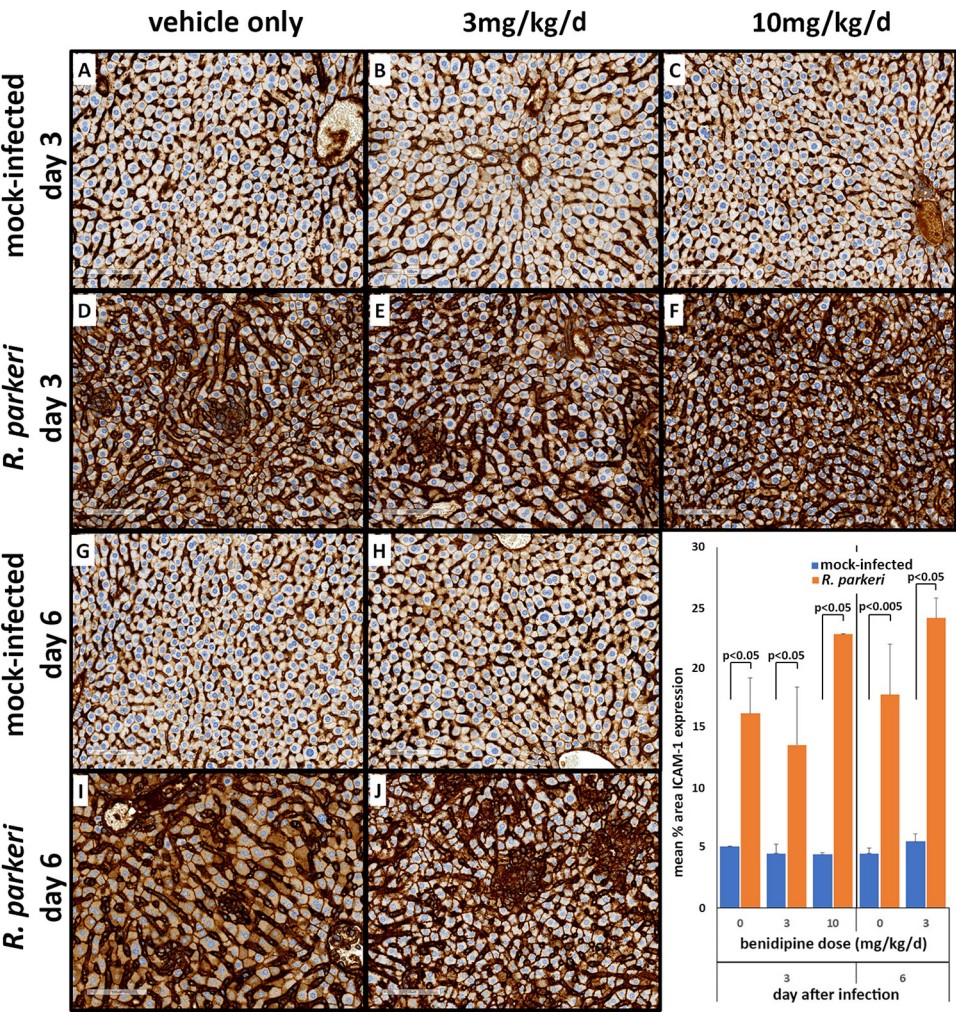

**Fig 6.** Comparison of immunohistochemical detection of ICAM1 on liver sections from no treated (A, D, G, I), treated with 3mg/kg (B, E, H, J), and 10mg/kg (C, F) of benidipine among infected and uninfected animals on day 3 and 6 p.i. Note the extensive upregulated expression with *R. parkeri* infection, regardless of benidipine treatment. Bar = 100 μm. Insert–image analysis quantification of ICAM-1 expression.

copies/mL copies in benidipine 3 mg/kg/d and 1 x $10^4$ *sca0* copies/mL copies in benidipine 10 mg/kg/d treated mice at day 3.

**_R. parkeri_ IHC** SFG rickettsiae IHC in liver (**Fig 8**) and spleen (**Fig 9**) paralleled the qPCR results (**Fig 7**). In the absence of benidipine, the distribution of *R. parkeri* in liver included localized infection in the hepatic lobular and periportal sinusoidal endothelial cells and mononuclear phagocytes, some of which were located within small inflammatory nodular infiltrates. The density of infected cells was modest on day 3 and very scant on day 6. In contrast, with benidipine use, *R. parkeri* was found in similar distributions, but also among large (approximately 50 μm diameter) inflammatory nodules at higher density at day 3, and as a suddenly severe infection extensively involving inflammatory infiltrates in hepatic lobular sinusoids and cells within vascular spaces at day 6. With infection in the presence of 10 mg/kg/d benidipine at day 3, the density of infected cells within hepatic lobular, perivascular and portal vein endothelial cells and mononuclear cells was higher than with no drug or 3 mg/kg/d, but the lack of inflammatory infiltrates in those treated with 10 mg/kg/d was striking.

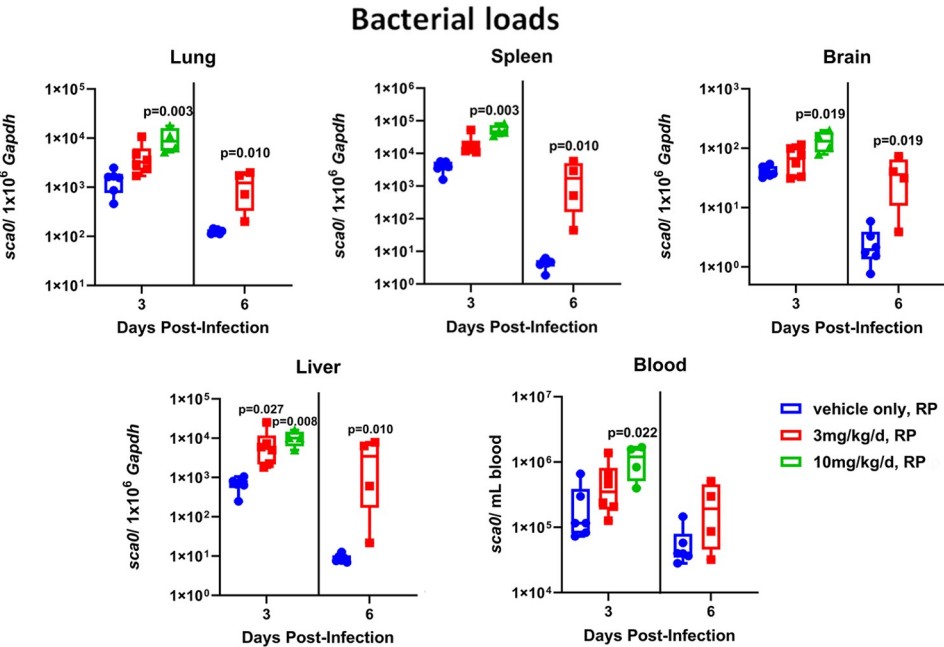

**Fig 7.** Measurement of rickettsial loads in lung (A), spleen (B), brain (C), liver (D), and blood (E) expressed as copies of *sca0* (*ompA*) per $10^6$ copies of mouse *Gapdh* in tissue or per milliliter of blood on days 3 and 6 p.i. of mice infected with 5 x$10^7$ viable *R. parkeri* (RP) organisms. Benidipine (3 or 10 mg/kg/d) administration led to a dose-dependent increase in *R. parkeri* load in lung, spleen, brain, liver and blood, compared to untreated (vehicle only) mice. Median rank is indicated by the line in the box, and the box boundaries show the 1st and 3rd quartiles, whereas the bars show the minimum and maximum ranks. P values are compared to untreated controls for each day.

In the spleen (**Fig 9**), in the absence of benidipine, *R. parkeri* IHC demonstrated moderately heavy infection almost exclusively in the red pulp on day 3 and a modest infection level on day 6, at which time the boundaries between the white and red pulp were nearly indistinguishable. *R. parkeri* was primarily found in sinusoidal endothelial cells and mononuclear phagocytes. With benidipine treatment at 3 mg/kg/d on day 3, there was a high level of infection, with numerous infected cells primarily within a moderately diminished red pulp, but also within cells in the white pulp including the follicles, the marginal zone and in the TCZ/PALS. Infected cells in the red pulp included endothelial cells and apparent mononuclear phagocytes. In the white pulp, *R. parkeri* appeared to be present in both endothelial cells and mononuclear phagocytes. By day 6, the infection was extensive, comprising possibly 10–20% of all cells, largely in the remaining red pulp and in the white pulp follicles, marginal zones and TCZ/PALS. With *R. parkeri* infection and 10 mg/kg/d benidipine at day 3, the infectious burden was as extensive as at day 6 with the lower dose, and involved primarily the red pulp, but also included cells within the white pulp follicles.

In regions of heavy infection, extensive inflammatory cell and necrotic cellular debris colocalized with infected cells.

## Cytokine analysis

Cytokine analyses on the acute phase serum revealed considerable changes in the expression of proinflammatory cytokines and chemokine directly related to *R. parkeri* infection, including IL6, IFNγ, TNFα, CXCL1 (KC), CCL2 (MCP1), and CCL5 (RANTES). Testing for IL12p70 failed and is not reported. The use of benidipine in either 3 or 10 mg/kg/d IP doses did not substantially alter cytokine expression beyond that of infection alone (**Fig 10**).

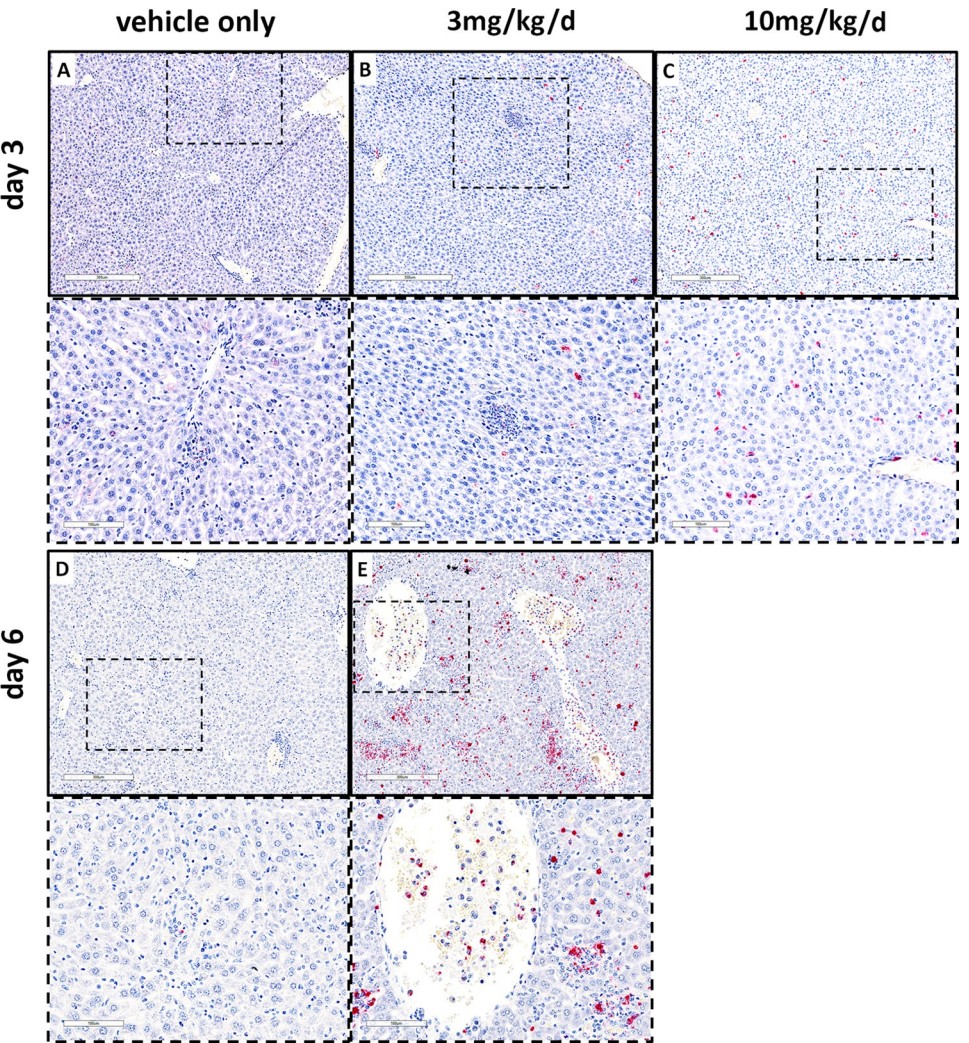

**Fig 8.** Comparison of immunohistochemical detection of SFG rickettsiae in liver: rickettsiae (red) increased with benidipine treatment (B, C, E) compared with the vehicle-treated control group (A, D). Frames with dashes represent a magnified area from image immediately above. Note the increased bacterial loads within the hepatic lobules related to benidipine dose and day of infection. Top images for each day bar = 300 μm; inserts (bottom row for each day) bar = 100 μm.

## Suppression of innate immune activation by benidipine in vitro

N8-3C3 NKT hybridoma cells were activated after loading rickettsial antigen, presumably *R. parkeri* glycolipids, on L-CD1d cells but not L-vector cells that do not express CD1d, as reflected by IL-2 production. NKT cells produced IL-2 levels lower than the limit of detection (1 pg/mL) when-co-cultured with L-vector control cells regardless of the presence of RP stimulus. L-CD1d endogenous lipids alone activate NKT cells to produce IL-2, and the addition of RP antigen enhanced NKT cell IL-2 production. The addition of 10μM benidipine significantly decreased NKT cell activation and IL-2 production of both the endogenous glycolipid response and that of the rickettsial NKT antigen. Data were analyzed using an one-way ANOVA. (**Fig 11**).

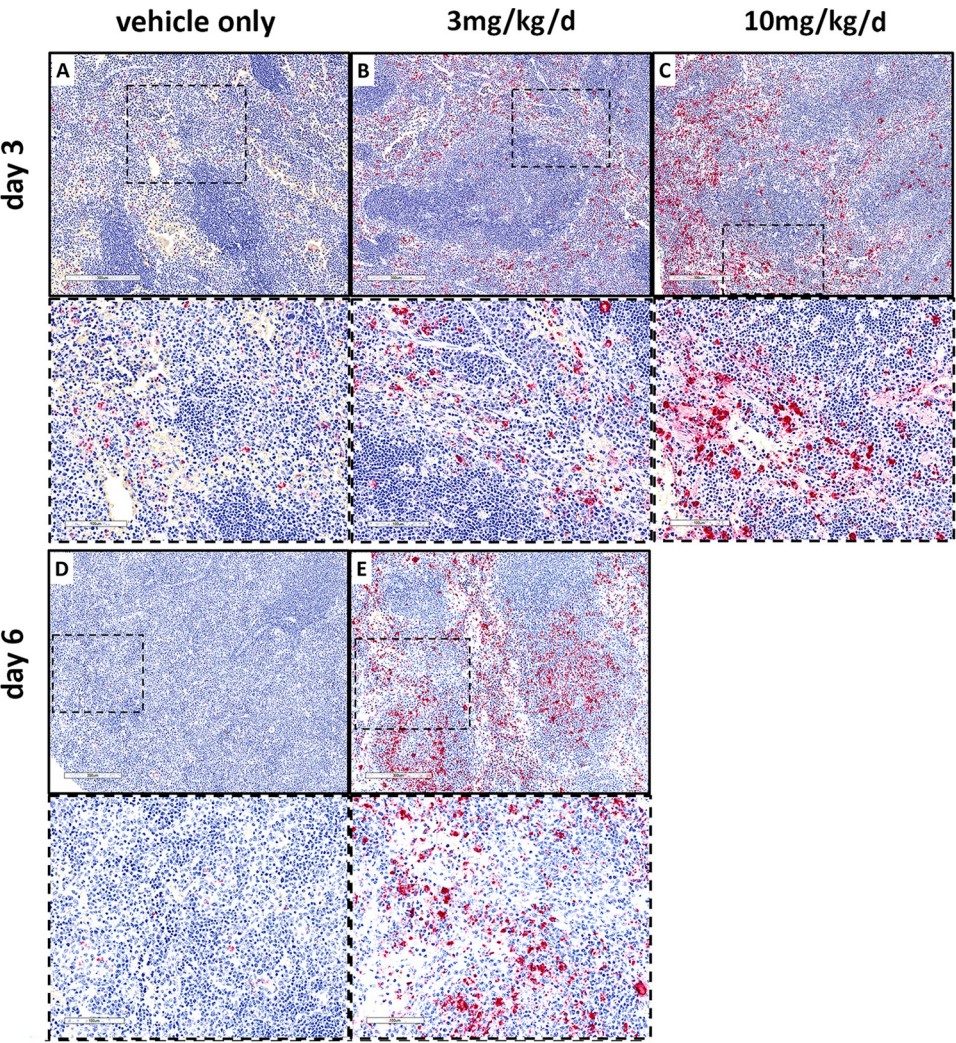

**Fig 9.** *R. parkeri* IHC (red) in spleen of benidipine-treated (B,C,E) and untreated (A,D) infected mice. Low and high magnifications shown for each day. Top images for each day, bar = 300 μm; inserts (bottom row for each day) bar = 100 μm. Note the dramatic increase in infection of splenic red pulp cell in animals treated with benidipine on day 3 (A, B, C) and extension of infection into white pulp as well on day 6 (D, E) compared to untreated.

## Discussion and Conclusions

The major pathophysiologic consequence of rickettsial infection is increased vascular permeability that leads to hypovolemia, hypotension, organ ischemia and septic shock [1]. The accompanying edema is a major factor in acute respiratory distress syndrome and brainstem herniation that can lead to death [27, 28]. Severity and untoward outcomes with rickettsial infections are believed to include factors related to rickettsial species or strains, and to host factors such as the production of proinflammatory cytokines that lead to increased permeability of microvascular endothelial cell barriers [1]. Preservation of low or normal intracellular calcium concentration $[Ca^{2+}]_i$ using calcium chelators, calcium channel blockers, or deletion of genes that express store-operated cation channels (SOC) abrogate vascular permeability after a stimulus with pathogens or thrombin [5, 29–32]. Given this and the important role that calcium plays in the pathophysiology of vascular permeability [6], we investigated drugs able to modulate calcium flux for their capacity to stabilize or reverse vascular permeability of *in vitro*

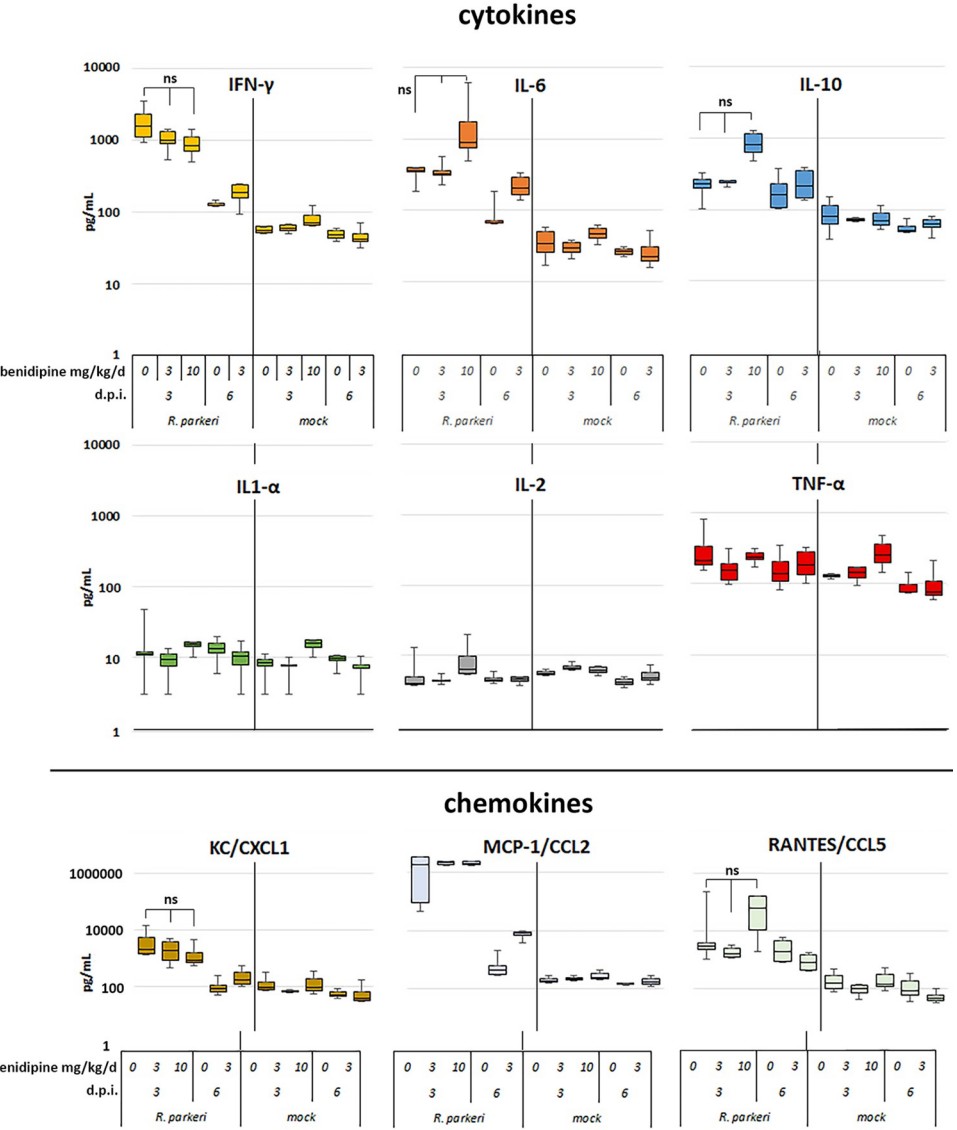

**Fig 10. Plasma cytokine and chemokine concentrations after *R. parkeri* or mock-infection and treatment ± 3 or 10 mg/kg benidipine.** Dramatic increases in concentrations are directly related to infection and are minimally modified by benidipine. Non-parametric t-tests with Bonferroni correction. While infection increased IFNγ, IL6, IL10, and all chemokines compared to mock-infected animals treated or not with benidipine, statistically significant changes comparing infected animals without benidipine to infected animals which received either 3 or 10 mg/kg/d benidipine were not observed, noted here by "ns". Median values are the central line in the box that represents the 1st and 3rd quartiles; whiskers show 0 and 4th quartiles (min and max values).

microvascular endothelial cell barriers infected by *R. parkeri* [29]. This led to the identification that benidipine, a dihydropyridine, L-, T-, and N-type voltage-gated calcium channel (VGCC) blocker in clinical use for hypertension, abrogates rickettsia-induced changes in transendothelial cell barrier electrical resistivity *in vitro* and could have potential uses *in vivo* as a tool to investigate the mechanisms of rickettsia-induced barrier deterioration or as a therapeutic [14,15]. Here, we describe modest improvement in vascular permeability and reduction in histopathologic inflammation in *R. parkeri*-infected mice treated simultaneously with benidipine,

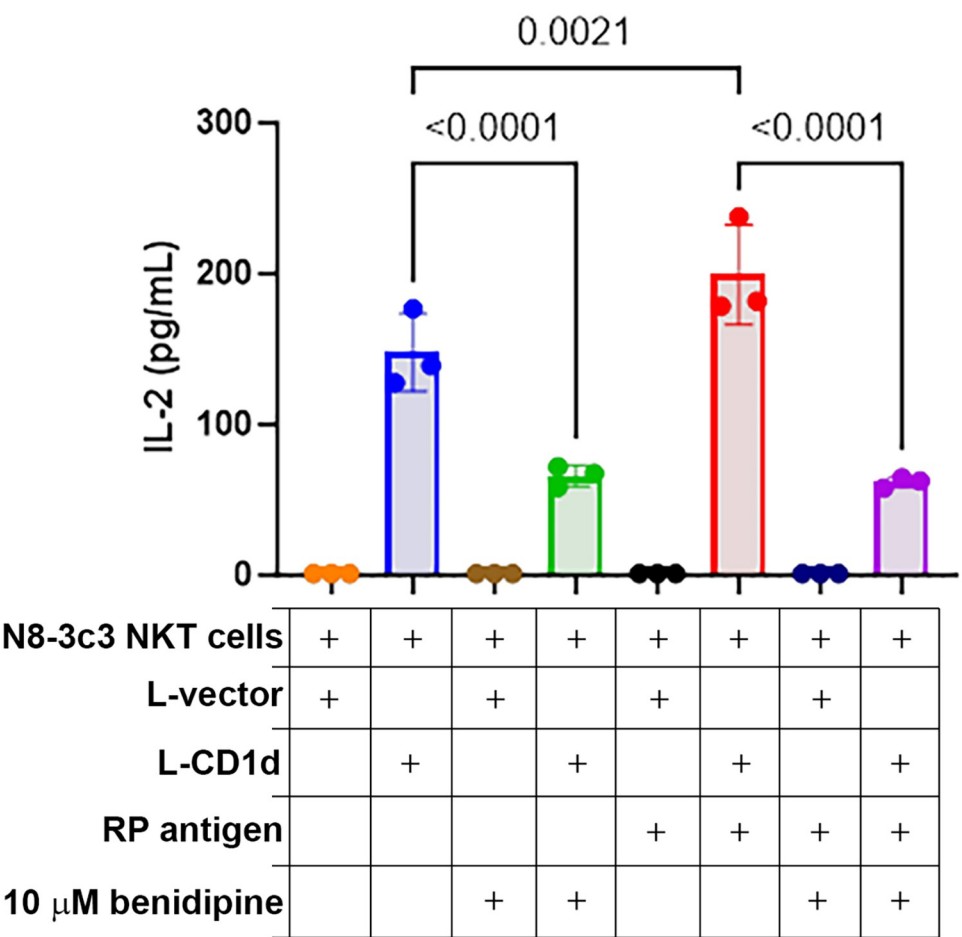

| | | | | | | | | |
|---|---|---|---|---|---|---|---|---|
| **N8-3c3 NKT cells** | + | + | + | + | + | + | + | + |
| **L-vector** | + | | + | | + | | + | |
| **L-CD1d** | | + | | + | | + | | + |
| **RP antigen** | | | | | + | + | + | + |
| **10 μM benidipine** | | | + | + | | | + | + |

**Fig 11. Treatment with 10μM benidipine decreases N8-3C3 NKT hybridoma cell activation and IL-2 production.**
IL-2 levels were measures by ELISA. NKT cells were not activated and produced IL-2 lower than the limit of detection
when co-culture with L-vector controls. L-CD1d endogenous lipids activate NKT cells to produce IL-2, and the
addition of RP antigen enhances NKT cell activation and IL-2 production. The addition of 10μM benidipine
significantly decreased NKT cell activation and IL-2 production. Data was analyzed using an ordinary one-way
ANOVA.

a finding that suggests a potential benefit to targeting calcium flux as an adjunct to control vascular permeability *in vivo*.

However, in the context of *R. parkeri* murine model of spotted fever rickettsiosis, use of benidipine at concentrations that had no apparent effect on uninfected mice resulted in negative impacts on antimicrobial control of rickettsiae and accelerated death, outcomes outweighing any benefit of benidipine-related reduced vascular permeability. The loss of antimicrobial function occurred possibly through the action of the drug on innate immune cells despite similar degrees of inflammatory histopathology in lungs, liver and brains of untreated animals.

An extensive literature search to determine appropriate benidipine dosing did not identify adverse consequences in mice, but a surprising result was the lethal toxicity of the highest dose regardless of rickettsial infection, which precluded the use of that dose in this study. Of greater interest was the dramatic expansion of *R. parkeri* tissue and blood loads with the two lower dose benidipine treatments–an unanticipated, previously undescribed response in an established murine model of human rickettsiosis in which mice received what was otherwise considered to be a safe pharmacologic agent.

Given known roles of calcium signaling in immune cell function [33–36], we spent considerable effort to characterize the how calcium channel blockers could lead to a marked adverse outcome *in vivo* while still reducing vascular permeability. Key findings of infection in the absence and presence of benidipine relate to i) suppression of weight loss and splenomegaly with treatment; ii) marked increases in pathogen blood and tissue load among infected benidipine-treated animals, as demonstrated by both PCR and IHC; iii) similar degrees of inflammatory histopathology in lungs, brains and livers in benidipine-treated and–untreated animals despite marked differences in pathogen loads; and iv) similar inflammatory cytokine and chemokine serum concentrations regardless of benidipine treatment.

The histopathologic findings offer insight into the mechanisms, despite somewhat similar appearances with infection in benidipine-treated and untreated animals. For example, animals treated with the higher benidipine dose (10 mg/kg/d) had significantly less lobular hepatitis and perivascular inflammatory infiltrates in liver despite significantly higher bacterial loads suggesting a defect in both inflammatory and innate immune responses that are mobilized in the absence of benidipine and that presumably initiate rickettsia immune control. Additionally, hepatic histopathology on day 6 in treated mice demonstrated significantly worse inflammation associated with lobular nodular aggregates of mononuclear cells and numerous neutrophils admixed with necrotic and apoptotic cellular debris, and portal vein thromboses. In contrast, such nodular aggregates were replaced by granuloma-like resolving lesions in untreated animals at the same time point. This observation suggests that the initial loss of innate immune function with benidipine treatment also leads to the inability to mature the antimicrobial protective and reparative responses that lead to diminishing rickettsial loads and resolving inflammatory histopathology.

Perhaps the most striking changes were in the spleen—the systemic immune organ examined in this study. The spleen is functionally divided into the red pulp—which provides a location for removal of senescent materials and surveillance for microbial infection—and the white pulp. The latter is divided into 3 major regions, the follicles, marginal zones, and T cell zone/periarteriolar lymphoid sheath (TCZ/PALS) [37,38]. Coordination of events between the red pulp and the white pulp zones are critical to functional antimicrobial events and induction of early innate and subsequent adaptive immune events. Much of the latter is driven by the capacity of cells within these zones upon activation to generate chemokine gradients and on their expression of unique targeting chemokine receptors that move cells from compartment to compartment in an organized fashion to orchestrate effective immune responses [39]. In fact, intracellular infection with the rickettsia *Ehrlichia muris*, in an animal model deficient in TNFα, leads to disturbances in follicular structure and germinal center formation because of the inability to express the chemokine CXCL13 that attracts B lymphocytes expressing CCR5 [40].

With *R. parkeri* infection and no benidipine, an initial rapid increase in cellular content is observed in the red pulp, in part related to the proliferation or relocation of CD3, CD4, CD8, and NK/NKT cells. Simultaneously, after a modest increase in follicle size with infection at day 3 related to slight increases in CD8 and NK/NKT cells, a dramatic reduction in cellularity of follicles, marginal zones and TCZ/PALS is observed by day 6 and is largely the result of the loss or migration of CD3 and CD4 lymphocytes, perhaps into the red pulp. The histopathologic context for these events is the initial fusion and early degeneration of follicular structure at day 3, followed by loss of architecture and fusion with red pulp, and the dramatic reduction in the presence of rickettsiae in the red pulp by day 6. In contrast, in the presence of benidipine, many of these events were abrogated, including the expansion and increasing cellularity of the red pulp, and loss of CD3 and CD4 cells from the white pulp follicles and TCZ/PALS. The lack of any alteration, including expansion of follicles with B cells regardless of treatment, is unclear, but could be the general result of the lack of significant TNFα production and its stimulation of CXCL13 expression in response to *R. parkeri* infection [40]. In general, the appearance is that of innate immune

anergy in that cells which proliferated or were mobilized by infection in animals without benidipine treatment, but became unresponsive in its presence. While some innate immune and inflammatory functions occurred, these data show that benidipine was associated with a significant decrement in antimicrobial activity compared to untreated animals and failure to organize the early reparative responses after microbial control on day 6 also as compared to untreated mice. This observation underscores the critical role for calcium, and calcium channels such as T, N, or L-type VGCCs impacted by benidipine, in the activation of protective and antimicrobial innate and early lymphocyte-mediated immune responses with spotted fever group rickettsiosis as modeled here by *R. parkeri* infection. To discern whether benidipine has the capacity to blunt or abrogate immune cell activation, we studied innate immune NKT cell hybridomas, in part based on the recognition of NKT and NK cell functions in another C3H/HeN murine model of spotted fever rickettsiosis (*R. conorii* Malish 7) toward the active control of severe infection [41–43]. Focusing on the early innate response that does not require prior adaptive immune stimulation as with CD4 and CD8 T lymphocyte responses, the results here demonstrate that lysates of purified *R. parkeri* contain ligands that can drive iNKT cell activation through CD1d-TCR recognition, and that this activation is markedly suppressed in the presence of benidipine. Whether similar responses occur with CD4 and CD8 T lymphocytes is not known, but likely given the requirement for calcium signaling for their activation as well [33]. These data are consistent with, but not conclusively supportive of the proposed mechanism owing to the complex integration of an array of signals required for immune response coordination.

Although the roles of L- and T-type VGCC targeted by benidipine and similar dihydropyridine class drugs in lymphocyte and T cell activation are not well understood, it is unanimously agreed that $Ca^{2+}$-mediated responses are critical for normal activation of some, but not all, immune cell functions and for cell-specific differentiation [33]. This includes the capacity to regulate activation at the TCR, through pathogen-associated molecular pattern ligand stimulation of toll-like receptors, alterations of $Ca^{2+}$ microdomains near $Ca^{2+}$-activated effector proteins, and even promotion of NFAT transcription factor nuclear entry [33,44,45]. As a result, a range of studies suggest the use of calcium channel blockers as immunomodulators for diseases that are chronic, and largely inflammatory or autoimmune [33,34,46], or for viral infections where the repurposed drugs may have distinct anti-viral activities [47–50]. In the case of severe acute rickettsiosis, the infectious process continues unabated with benidipine and further exacerbates inflammatory response and loss of antimicrobial activity, apparently in part owing to the loss of immune cell activation.

Interestingly, these observations show that bacterial burden and inflammatory cell infiltration are not direct correlates of increased vascular permeability and survival during spotted fever group rickettsiosis, but provide no real confidence that separate pharmacologic targeting of vascular permeability could be possible, at least with benidipine. That benidipine did not alter expression of a key leukocyte adhesion molecule, ICAM1, or the production of a range of important inflammatory cytokines and chemokines, shows that not all of the processes integrated into an antimicrobial inflammatory response depend on calcium signaling, at least at the levels examined here. This underscores the potential multiplicity of responses triggering redundant functions but through distinct mechanisms, that without coordination from activated and migrating immune cells within the immune architecture of the spleen or lymph nodes, can lead to severe uncontrolled inflammatory disease without effective antimicrobial activity. The finding of similar chemokine expression profiles with or without benidipine during *R. parkeri* infection seems contradictory to the requirement for specific chemokines and their cognate receptors to coordinate immune response and immune cell migration in the spleen and lymph nodes. However, the chemokines studied here were selected because of their role in inflammatory reactions, not immune cell migratory coordination. While not studied here, the assessment

of chemokines most critical for orchestrating the migration of cells during immune responses will further help define the potential migratory events in rickettsial innate immune response.

Overall, these findings support the role of calcium as an essential second messenger to regulate and coordinate intracellular signaling pathways that play important roles in immune cells in relation to selection, development, proliferation, activation, cytotoxic function, production of antimicrobial effectors, phagocytosis, and phagosome-lysosome fusion, among a range of other functions [34,51]. The explanation for these observations requires considerable further study, including careful analyses that examine attributes of individual immune cells, their activation markers and functional assays. Current data suggest that the key cells that initiate innate host immunoprotective responses and promote resolution of severe acute inflammatory responses are inhibited or modulated negatively by benidipine and perhaps other commonly used calcium channel blockers. This very likely diminishes the full range of calcium activation signals and the capacity to activate critical immune cells in the generation of well-coordinated and protective anti-rickettsial responses. It would be of great interest to study severity of human rickettsial disease through the lens of calcium channel use to determine if the severe disease phenotype observed here occurs and contributes to morbidity with spotted fever group rickettsioses, and whether cell-specific targeting of calcium-related signals could promote stronger anti-rickettsial innate immune responses or control of vascular permeability.

## Supporting information

**S1 Fig.** High resolution image comparison of hepatic pathology on day 6 after *R. parkeri* infection with no benidipine (left panels) and with 3 mg/kg/d benidipine (right panels). The top left H&E stain panel demonstrates the waning lobular and periportal inflammatory infiltrates with focal portal vein thrombosis (green arrow) and small granuloma-like aggregates (yellow arrows) in untreated infected animals, whereas the infected treated animals were more likely to have persistent lobular aggregates (white arrows) of inflammatory cells that included both mononuclear cells and neutrophils, as well as necrotic and apoptotic cells (white arrowheads and insert) and more frequent portal vein thromboses (right panel, green arrows). The bottom panels demonstrate the rare presence of *R. parkeri* by IHC (Rc7 IHC) in the livers of infected untreated animals (bottom left), but an inability to control the infection in animals treated with benidipine. Bar = 90 μm.
(TIF)

**S2 Fig.** High resolution H&E-stained images of liver from *R. parkeri*-infected mice treated with vehicle only, or with 3 or 10 mg/kg/d benidipine. Note the lobular hepatitis with nodular mononuclear cell inflammatory infiltrates of similar distribution and intensity in untreated and mice treated with 3 mg/kg/d benidipine on day 3 and worse on day 6. Note also the diminished lobular hepatitis on day 3 in animals treated with 10 mg/kg/d benidipine. The inserts demonstrate the differences in inflammatory cell content in the nodular infiltrates in the hepatic lobules where treatment with benidipine results in the supplemental recruitment of neutrophils with greater tissue necrosis compared to untreated infected mice.
(TIF)

**S3 Fig.** High resolution H&E-stained images of spleen from *R. parkeri*-infected mice treated with vehicle only, or with 3 or 10 mg/kg/d benidipine. Note the retained by fragmenting follicles and white pulp with increasing cellularity in the red pulp of infected untreated animals at day 3, followed by the extensive loss of white matter and follicular architecture and massive expansion of red pulp by mononuclear cells in untreated animals at day 6. The red arrows and insert demonstrate small neutrophilic and necrotic regions within the red pulp at day 3, and

the resolution of these foci by reparative granuloma-like foci (white arrows and insert) at day 6 when *R. parkeri* burden is at a nadir by PCR and IHC. In contrast, infected animals treated with benidipine had significantly less red pulp expansion and worse fragmentation of white pulp and follicles with clear evidence of substantial neutrophilic infiltration and tissue/cellular necrosis (red arrows 3 mg/kg/d benidipine, blue arrows 10 mg/kg/d benidipine, and inserts). Also note the extensive phagocytosis of cellular and nuclear debris in macrophages within follicles with benidipine (insert day 3, benidipine 10 mg/kg/d). By day 6 with animals receiving 3 mg/kg/d benidipine, there was extensive red pulp necrosis (red arrows and insert) lacking any evidence of tissue repair at a time when *R. parkeri* burden in the spleen is at peak.
(TIF)

**S4 Fig.** Representative high resolution H&E stained image of lung from and *R. parkeri*-infected mouse treated with vehicle only (left panel) or treated (right panel) with 3 mg/kg/d benidipine and necropsied on day 6 p.i. Note the dense mixed inflammatory cell interstitial infiltrates that widen alveolar septa, the segmental vasculitis of pulmonary venules (black arrows), capillaritis with karyorrhectic debris (red arrows), and focal alveolar wall necrosis in the treated animal (green arrow).
(TIF)

**S5 Fig. High resolution image visualizations of fluorescent dextran vascular leakage as a measure of vascular permeability with and without benidipine.** The vascular leakage images that were at the median-rank for each tissue, time p.i., infection or mock infection, and drug dose are shown. For dextran extravasation measurements from liver and brain, the percent area of fluorescence was measured over the entire image. For the lungs, large vascular structures were excluded from the fluorescent images to offset intravascular retention fluorescent signal. Note the minor variations over time, infection status and drug treatment, none of which were reproducibly significant.
(TIF)

**S6 Fig.** Representative CD3 IHC on splenic tissue from *R. parkeri*-infected and mock infected animals at days 3 and 6 d.p.i. with or without benidipine treatment as indicated. Note the highest density in uninfected animals within the white pulp and TCZ/PALS, and the marked increase in CD3 cells in the red pulp of infected animals. Bar = 500 μm.
(TIF)

**S7 Fig. Representative CD4 IHC on splenic tissue from *R. parkeri*-infected and mock infected animals at days 3 and 6 d.p.i. with or without benidipine treatment as indicated.** As for CD3 cells, note the predominant restriction of CD4 cells to the white pulp follicles and TCZ/PALS in uninfected animals, and the loss from the regions accompanied by a simultaneous increase in CD4 cells in the red pulp of infected animals. Note that benidipine treatment in infected mice resulted in less white pulp depletion of CD4 cells and a diffuse distribution of CD4 cells throughout splenic parenchyma compared to the clustered appearance in the red pulp with infection but no benidipine. Bar = 500 μm.
(TIF)

**S8 Fig. Representative CD8 IHC on splenic tissue from *R. parkeri*-infected and mock infected animals at days 3 and 6 d.p.i. with or without benidipine treatment as indicated.** Note the scant distribution of CD8 cells in the red pulp of uninfected animals compared to the marked increase in red pulp CD8 cells with infection, as well as the diffuse distribution, including within white pulp follicles, with benidipine treatment and infection. Bar = 500 μm.
(TIF)

**S9 Fig. Representative NCR1 (NK/NKT) IHC on splenic tissue from *R. parkeri*-infected and mock infected animals at days 3 and 6 d.p.i. with or without benidipine treatment as indicated.** Note the discrete localization of NCR1$^+$ cells within the red pulp of uninfected mice, and the marked increase in numbers with infection. With benidipine treatment in infected mice, the localization of NCR1$^+$ cells became more diffuse, including incursion into white pulp follicles and TCZ/PALS. Bar = 500 μm.
(TIF)

**S10 Fig. Representative CD45R/B220 (B cell) IHC on splenic tissue from *R. parkeri*-infected and mock infected animals at days 3 and 6 d.p.i. with or without benidipine treatment as indicated.** Note the localization of B220$^+$ cells primarily in marginal zones around the white pulp follicles in uninfected animals, and the more diffuse distribution into the red pulp with benidipine. Total quantity of B220$^+$ cells does not dramatically change through these conditions. Bar = 500 μm.
(TIF)

**S1 Raw Data. Raw data and statistical analyses are provided as data files and correspond to the data in each of the Figures cited in the text, except the histologic images for which the original data is presented as the image itself.** Fig 1. Relevant raw data and statistical calculations for data in Fig 1. Fig 1A, contains files relevant to the calculation of survival; Fig 1B contains raw data and statistical calculations relevant to change in body weight over time; Fig 1C contains files relevant to raw data of spleen weights and statistical calculations. Fig 2GHI. The excel file within provides raw data regarding the ranking of histopathology injury in lung, brain and liver, and the associated statistical evaluations. Fig 4. Relevant raw data from ImageJ analysis of dextran extravasation in liver, lung and brain, including statistical analyses. Fig 5. Raw data used to calculate semiquantitative estimates of specific pathologic features in the spleens of animals, weighted by spleen weight, including H&E histology, and immunohistochemistry analysis of immune cells and their microanatomical distributions. Fig 6. Relevant raw data from ImageJ analysis of ICAM1 immunohistochemical staining in liver, including statistical analyses. Fig 7. Fig 7ABC contains files with raw data of quantitative PCR in brain, lung, spleen, and statistical analyses from those data; Fig 7D contains files with raw data of qPCR in liver and statistical analysis from those data; Fig 7E contains file with raw qPCR data in blood and associated statistical analyses from those data. Fig 10. Relevant raw data from Bioplex cytokine and chemokine analyses of plasma. These data include those that were used for statistical calculations. Fig 11. This contains a file including raw data from the NKT activation experiments in the manuscript's figure 11. Included also are the statistical evaluations.
(ZIP)

## Acknowledgments

The authors would like to thank the staff of the NIAID Vaccine Research Center animal facility and IACUC for their support and kind assistance during the COVID-19 pandemic. Special thanks to Tonya Webb for providing the tools and reagents to study NKT cell activation.

## Disclaimer

The opinions expressed herein are those of the author(s) and are not necessarily representative of those of the Uniformed Services University of the Health Sciences (USUHS), the Department of Defense (DOD); or, the United States Army, Navy, or Air Force.

## Author Contributions

**Conceptualization:** Dennis J. Grab, J. Stephen Dumler.

**Formal analysis:** Andrés F. Londoño, Jennifer M. Farner, J. Stephen Dumler.

**Funding acquisition:** Dennis J. Grab, Diana G. Scorpio, J. Stephen Dumler.

**Investigation:** Andrés F. Londoño, Jennifer M. Farner, Marlon Dillon, Yuri Kim.

**Methodology:** Andrés F. Londoño, Jennifer M. Farner, Dennis J. Grab, J. Stephen Dumler.

**Project administration:** Andrés F. Londoño, Jennifer M. Farner, Diana G. Scorpio, J. Stephen Dumler.

**Resources:** Andrés F. Londoño, Jennifer M. Farner, Marlon Dillon, Diana G. Scorpio.

**Supervision:** Andrés F. Londoño, Diana G. Scorpio, J. Stephen Dumler.

**Validation:** Andrés F. Londoño.

**Visualization:** Andrés F. Londoño, Jennifer M. Farner, J. Stephen Dumler.

**Writing – original draft:** Andrés F. Londoño, Jennifer M. Farner.

**Writing – review & editing:** Andrés F. Londoño, Jennifer M. Farner, Dennis J. Grab, Yuri Kim, J. Stephen Dumler.

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
