## [Decision Letter · Decision Letter 0]

9 Oct 2023

Dear Dr. Dumler,

Thank you very much for submitting your manuscript "Benidipine stabilizes vascular permeability but impairs innate immunity in a murine model of spotted fever rickettsiosis" for consideration at PLOS Neglected Tropical Diseases. As with all papers reviewed by the journal, your manuscript was reviewed by members of the editorial board and by several independent reviewers. In light of the reviews (below this email), we would like to invite the resubmission of a significantly-revised version that takes into account the reviewers' comments. 

We cannot make any decision about publication until we have seen the revised manuscript and your response to the reviewers' comments. Your revised manuscript is also likely to be sent to reviewers for further evaluation.

Sincerely,

Bin Gong, Ph.D., M.D.

Guest Editor

Joseph Vinetz

Section Editor

Reviewer's Responses to Questions

**Key Review Criteria Required for Acceptance?**

**Methods**

-Are the objectives of the study clearly articulated with a clear testable hypothesis stated?

-Is the study design appropriate to address the stated objectives?

-Is the population clearly described and appropriate for the hypothesis being tested?

-Is the sample size sufficient to ensure adequate power to address the hypothesis being tested?

-Were correct statistical analysis used to support conclusions?

-Are there concerns about ethical or regulatory requirements being met?

Reviewer #1: Line 164: The molecular weight of the dextran to be stated here.

Line 180: The word " preformed" does not make sense.

Reviewer #2: Please see below

Reviewer #3: Yes, the methods used in this study are appropriate and statistical analyses were performed as necessary.

**Results**

-Does the analysis presented match the analysis plan?

-Are the results clearly and completely presented?

-Are the figures (Tables, Images) of sufficient quality for clarity?

Reviewer #1: Lines 195 and 197: It does not make sense that animals were given a sublethal dose and became moribund

Line 223: It is more impressive that the organ titers were increased than the rickettsemia.

Line 238 and throughout the results: The magnifications are insufficient to visualize the stated results. For example fibrinoid necrosis is not detectable at the magnification provided. The stated results are not questioned, but the statements that a large number of the lesions are visible in the figures are incorrect.

Line 240: The statement is incorrect that meningitis is present in figure 3 C.

Figure 3G: Are there any statistical differences in these values?

Lines 255-259 and figure 5: Determination of vascular permeability in the liver is confounded by the discontinuous sinusoidal lining that includes both Kupffer cells and endothelial cells. Evaluation of vascular permeability in an organ with endothelial tight junctions such as the brain would be preferred.

Legend of figure 5. Hepatic loads were not determined here but rather fluorescence.

Lines 260 and 262: "a modest potential therapeutic window" is stated redundantly.

Figure 6: No red arrows are visible.

Line 313 and throughout the manuscript: The word fulminant indicates a rapid time of development. Its use to mean increased severity or mortality is incorrect.

Line 297: figures S3-S7 do not all demonstrate CD4 cells as the sentence states.

Line 692: figure S8 was not provided in the manuscript.

Reviewer #2: Please see below

Reviewer #3: Yes, the obtained results were presented well.

**Conclusions**

-Are the conclusions supported by the data presented?

-Are the limitations of analysis clearly described?

-Do the authors discuss how these data can be helpful to advance our understanding of the topic under study?

-Is public health relevance addressed?

Reviewer #1: Line 355: It is unclear what is meant by "store-operated".

Reviewer #2: Please see below

Reviewer #3: I don't think the data presented fully support the claims including the title. Please see the general comments.

**Editorial and Data Presentation Modifications?**

Reviewer #1: (No Response)

Reviewer #2: Please see below

Reviewer #3: NA

**Summary and General Comments**

Reviewer #1: This manuscript is an excellent contribution to the field of knowledge. In addition to many minor flaws, the discrepancy between what was observed and stated in the manuscript compared to what was visible in the figures is striking. This needs to be addressed effectively.The authors should recognize what features such as apoptosis and polymorphonuclear leukocytes are not visible in the figures.

Reviewer #2: The manuscript explores the repurposing of benidipine, a calcium channel blocker, to diminish the vascular dysfunction caused by Rickettsia parkeri in vivo in an experimental animal model of spotted fever rickettsiosis. Animals were treated with a sub-lethal dose of bacteria, followed by i.p. administration of different doses of benidipine. 

Even though the manuscript described a decrease in vascular permeability, overall, the drug is not capable of reversing the effects of R. parkeri and in some parameters, the effects are enhanced after combining the drug with the bacteria. Likewise, the control group of uninfected mice treated with benidipine, displayed deleterious symptoms on the immune response (increased cellularity in splenic tissue of mice treated with 10 mg/kg/d of benidipine alone and release of inflammatory mediators) that debunk the primary hypothesis. The authors should not suggest the usage of this drug in infections by R. parkeri due to the adverse reactions it could cause. 

I consider the researchers needed to perform a pilot study of the drug itself, before conducting the actual experiments with the pathogenic bacteria, since the doses employed affected the groups in such a manner that the audience could question the whole experimental design. Regardless of the works cited by the authors to support the doses used, the strain of mice employed in this study is different as well as the drug’s administration route, compare to the studies repurposing the use of this drug for Chagas disease and estimating its acute toxicity, respectively, which could affect the final response seen in the controls.

In addition, the manuscript must be revised. It would be interesting to display the data of the mice uninfected but treated with 10/mg/kg/d of benidipine to see the alterations caused by the drug itself. Likewise, legends must be summarized and the discussion must be revised since the paragraphs are too long and key references are missing to support the ideas explained by the authors.

Reviewer #3: This article from Londono and colleagues report the unexpected impact of benidipine in impairing innate immunity while controlling the vascular permeability in a mouse model of spotted fever rickettsiosis. This is well written article and presents elegant data to demonstrate how benidipine exacerbates the infection by increasing bacterial load and altering the tissue morphology. However, the mechanistic details to demonstrate the effects of benidipine in affecting innate immunity specifically via calcium signalling is missing. Most of the results only present the actual obeservations in the animal model, not the actual mechanistic details to claim that benidipine is specifically affecting the innate immunity, possibly via intracellular calcium signalling. So I would strongly suggest the authors to perform some additional experiments to demonstrate the direct effect of benidipine on immune cells in the presence and absence of R.parkeri. This could be done in culture cells if using animal models is tedious. If it is not possible to perform these additional experiments, then the authors should thoroughly edit this manuscript just to present the observational data and highlight the need for further studies in the discussion section to underpin the mechanistic details. They should also consider changing the title. 

Some additional comments to consider while revising this manuscript. 

1. There are some typographical errors in the introduction and discussion sections. Please correct them thoroughly. 

2. Some figures are presented in excel quality, which don't look great. So I would strongly suggest the authors to prepare the final figures using another tool, and label all the axes clearly for all data. 

3. The figure legends are repeating the texts from results section. Instead, please only highlight the brief methods used to generate the data, and statistical analysis. There is no need to repeat the results in legends. 

4. In Figure 1B, please explain the missing line for 10 mg/kg in the legend. It would be helpful to show the statistical significance on this image somehow. For you could create a box/bar diagram for important data points. 

5. I am bit lost with the data presented in figure 3 because of the ranking system. I read several times, but couldn't completely understand the ranking system used. So please simply the text here and better explain this. Or the data can be presented similar to those in figure 4 and use the ranking system for the quantification but in a simple format. 

6. Figure 5 - please create a better figure for quantified data, and include multiple comparisons between different panels. 

7. Figure 6 - panel B is really confusing. again, please use high quality figures and justify the staining for different markers. 

8. Figure 8 and 9 - please see if you can provide some form of quantified data here. 

9. Figure 10 - the data on cytokines and chemokines need better explanation. It's not clear how the statistical analyses were performed. I would suggest comparing all treated data with untreated controls and check the significance. Moreover please justify the use of these specific markers better. More mechanistic details are required for IL6, IL10 and RANTES as these were largely increased by benidipine. 

10. Did the authors measure the blood parameters in these mice after treating with the drug? If yes, they will also demonstrate more mechanistic details about the impact of this drug. 

11. What sort of pathology was observed in dead mice due to high doses of benidipine? This will also add more to this study.

PLOS authors have the option to publish the peer review history of their article (what does this mean?). If published, this will include your full peer review and any attached files.

Reviewer #1: No

Reviewer #2: No

Reviewer #3: No
---

## [Decision Letter · Decision Letter 1]

30 Jan 2024

Dear Dr. Dumler,

Thank you very much for submitting your manuscript "Benidipine impairs innate immunity converting sublethal to lethal infections in a murine model of spotted fever rickettsiosis" for consideration at PLOS Neglected Tropical Diseases. As with all papers reviewed by the journal, your manuscript was reviewed by members of the editorial board and by several independent reviewers. The reviewers appreciated the attention to an important topic. Based on the reviews, we are likely to accept this manuscript for publication, providing that you modify the manuscript according to the review recommendations. 

Sincerely,

Bin Gong, Ph.D., M.D.

Guest Editor

Joseph Vinetz

Section Editor

Reviewer's Responses to Questions

**Key Review Criteria Required for Acceptance?**

**Methods**

-Are the objectives of the study clearly articulated with a clear testable hypothesis stated?

-Is the study design appropriate to address the stated objectives?

-Is the population clearly described and appropriate for the hypothesis being tested?

-Is the sample size sufficient to ensure adequate power to address the hypothesis being tested?

-Were correct statistical analysis used to support conclusions?

-Are there concerns about ethical or regulatory requirements being met?

Reviewer #1: No issues

Reviewer #2: The authors have improved the method section, however, I recommend to revise for minor details: 

Line 122: add the units for the amount of cell-free bacteria that was inoculated.

Revise the subsection about "Suppression of innate immune activation by benidipine in vitro". I had to read it several times to understand it.

Line 213: What hBMECs stands for?

Reviewer #3: Yes

**Results**

-Does the analysis presented match the analysis plan?

-Are the results clearly and completely presented?

-Are the figures (Tables, Images) of sufficient quality for clarity?

Reviewer #1: The interpretation of the data that the levels of cytokines and chemokines are not significantly different with and without benidipine treatment does not seem to conform to what is shown in the figure, which shows increased gamma interferon and IL-6 with treatment. The *and ** are stated in the figure legend to indicate p<0.005 (is p<0.05 what was really meant?) and p<0.001. These indeed would be significantly different.

Reviewer #2: The authors have improved the section, the new data and modification done to the figures allow a better understanding of the findings.

Reviewer #3: Yes for the texts, but I am still not happy with the excel quality figures. All the bar diagrams should be prepared in high quality without any gridlines, and show the statistical significance clearly.

**Conclusions**

-Are the conclusions supported by the data presented?

-Are the limitations of analysis clearly described?

-Do the authors discuss how these data can be helpful to advance our understanding of the topic under study?

-Is public health relevance addressed?

Reviewer #1: (No Response)

Reviewer #2: The authors have improved the section, the new data and analysis added valuable information that support the conclusions.

Reviewer #3: Yes

**Editorial and Data Presentation Modifications?**

Reviewer #1: Revised manuscript would be improved for clarity by addressing the following:

1. Lines 44-46: It should be clarified that a retrospective review of patient's medications and outcomes are intended rather than a clinical trial.

2. Line 61-63: Pathogens should not be anthropomorphized. They have no intent.

3. Line 83: Does accelerated mean faster poor increased severity?

4. Lines 125-127: Please fix the grammar. It is unclear where the first independent clause ends.

5. Line 135: manufacturer's is the possessive case and requires an apostrophe.

6. Lines 157 and 177: Please change the measure of concentration, micromolar, to the appropriate measure for thickness, micrometer.

7. Line 193: The word performed would make sense rather than preformed.

8. Line 344: Is "reduction of white pulp" intended?

9. Lines 353 and 502: What is the meaning of granuloma-like. I do not identify groups of activated macrophages.

10. Line 430: Is "cultured" meant?

11. Line 431: Is the correct word present or presence?

12. Line 153: Perivascular edema occurs in the brain in spotted fever group rickettsioses, but brainstem herniation requires a cited reference to confirm that it occurs.

13. Line 522: aut versus and

Reviewer #2: (No Response)

Reviewer #3: I am not sure if the production team can improve the figures, if yes, then they can do this.

**Summary and General Comments**

Reviewer #1: The scientific rigor and contents of this manuscript are excellent.

Reviewer #2: The authors addressed the reviewers' main concerns and have improved the manuscript.

Reviewer #3: The authors have addressed most of my comments, and explained why they couldn't address some of the comments. Overall, the manuscript looks much better now. As mentioned above, I would suggest the authors to replace the Excel figures with high quality bar diagrams.

PLOS authors have the option to publish the peer review history of their article (what does this mean?). If published, this will include your full peer review and any attached files.

Reviewer #1: No

Reviewer #2: No

Reviewer #3: No

Figure Files:

Data Requirements:

Reproducibility:

References

---

## [Editor Report · Decision Letter 2]

13 Feb 2024

Dear Dr. Dumler,

We are pleased to inform you that your manuscript 'Benidipine impairs innate immunity converting sublethal to lethal infections in a murine model of spotted fever rickettsiosis' has been provisionally accepted for publication in PLOS Neglected Tropical Diseases.

Best regards,

Bin Gong, Ph.D., M.D.

Guest Editor

Stuart Blacksell

Section Editor

---

## [Editor Report · Acceptance letter]

22 Feb 2024

Dear Dr. Dumler,

We are delighted to inform you that your manuscript, " Benidipine impairs innate immunity converting sublethal to lethal infections in a murine model of spotted fever rickettsiosis ," has been formally accepted for publication in PLOS Neglected Tropical Diseases.

Best regards,

Shaden Kamhawi

co-Editor-in-Chief

Paul Brindley

co-Editor-in-Chief
